# Assessing Regional Differences in Green Innovation Efficiency of Industrial Enterprises in China

**DOI:** 10.3390/ijerph16060940

**Published:** 2019-03-15

**Authors:** Jun-liang Du, Yong Liu, Wei-xue Diao

**Affiliations:** School of Business, Jiangnan University, Wuxi 21,4122, China; 6170903007@stu.jiangnan.edu.cn (J.-l.D.); 6180903007@stu.jiangnan.edu.cn (W.-x.D.)

**Keywords:** green technology innovation, R&D stage, transformation stage, efficiency measurement, difference, three wastes

## Abstract

Green technology innovation is an important means to break out of the constraints of resources and the environment, enhance the competitiveness of enterprises, and achieve the upgrading of industrial structures, and promote high-quality economic growth. In order to realize the overall improvement of the green technology innovation capability of Chinese enterprises, it is necessary to measure the efficiency of industrial enterprises’ green technology innovation and explore their regional differences. In this paper, from the perspective of a two-stage innovation value chain, by introducing the industrial carbon emissions per unit of Gross Domestic Product (GDP) and the “three wastes” pollutants into the research framework of green technology innovation efficiency, we established a novel green innovation efficiency evaluation indicator system for industrial enterprises. Furthermore, we used a two-stage network DEA with shared input to measure the efficiency of regional enterprises’ green technology innovation and explored the regional differences in industrial enterprises’ green technology R&D and the efficiency of green technology achievement transformation. Finally, we provide some suggestions for improving China’s industrial enterprises’ green innovation efficiency, so that they can ameliorate the significant regional imbalances and differences and realize high-quality economic growth.

## 1. Introduction

Severe climate change and environmental pollution problems have been attracting increasing attention from countries around the world. Transforming traditional economic growth patterns, effectively improving environment quality while maintaining original levels of economic development, and making good use of resources to achieve sustainable development have become important development initiatives for many countries. As an important means to get rid of resources and environmental constraints, enhance enterprise competitiveness, achieve industrial structure upgrading, and promote high-quality economic growth, green technology innovation is constantly accepted by more countries [1]. Innovation efficiency reflects the efficiency of a company’s use of innovative resources and has a profound impact on the formation of competitive advantage [2]. Facing serious resource and environmental problems, Brawn and Wield [3] proposed the concept of green technology innovation to improve national or regional competitiveness and achieve the healthy development of national or regional economy. Afterward, some scholars have studied green technology innovation efficiency from the perspectives of natural resources [4], institutional theory [5], social network relationships [6], stakeholders [7], and CSR (Corporate Social Responsibility) [8]. To sum up, these studies mainly focused on defining the definition of green technology innovation; analyzing the relationship between green technology innovation, environmental rules and social economics [9,10]; identifying the factors affecting green technology innovation [11,12,13]; constructing an evaluation index system for green technology innovation efficiency and measuring the efficiency of green technology innovation. In 2017, Thomson Reuters announced the list of the top 100 innovative enterprises in the world, however, the list did not include any mainland Chinese companies. It shows that the technological innovation output efficiency of industrial enterprises still needs to be improved in China.

In addition, with the growth of the Chinese industrial economy, the environmental pollution problem is becoming increasingly serious. The contradiction between technology-driven economic development and the ecological environment has become increasingly prominent, and people’s lives have been greatly affected. However, current studies on how to measure green technology innovation efficiency for China’s provincial industrial enterprises still have some shortcomings. For example, the current research does not fully consider the phased characteristics of green technology innovation efficiency and the relevance of shared inputs. The evaluation indicator system constructed is unreasonable, which makes it impossible to fully discover the differences and characteristics of statistical data. Therefore, a scientific and reasonable indicator system and measurement method are the basis for measuring the efficiency of green technology innovation. In order to better study the regional differences in green innovation efficiency of Chinese industrial enterprises, this paper establishes a model based on Chinese industrial enterprises. In this paper, with respect to the problems of regional differences in China’s industrial enterprises’ green innovation efficiency, from the perspective of a two-stage innovation value chain, taking China’s industrial enterprises as the research objects, we established a novel green innovation efficiency evaluation index system for industrial enterprises by introducing the industrial carbon emissions per unit of GDP and the “three wastes” pollutants into the research framework for green technology innovation efficiency. Then, this study used a two-stage network DEA with shared input to measure the efficiency of regional enterprises’ green technology innovation and explored the regional differences in industrial enterprises’ green technology R&D and green technology achievement transformation.

The content of this paper is organized as follows. In Section 2, we provide an overview of the related literature on green technology innovation. In Section 3, a novel green technology innovation efficiency evaluation index system of industrial enterprises is designed. Section 4 establishes a two-stage network DEA model with shared related input. In Section 5, we use the two-stage network DEA model to measure regional industrial enterprises’ green innovation efficiency and analyze their differences. Section 6 concludes the paper with some remarks and provides some suggestions.

## 2. Literature Review

According to the needs of ecological and economically balanced development, we must find a relationship among rapid economic development, excessive resource utilization, and natural environmental degradation. By measuring the efficiency of green technology innovation, we can find key influencing factors and promote the sustainable development of green ecological economy [14,15,16]. The measurement and evaluation of green technology innovation efficiency involves multi-stages, multi-angles, and multi-factors, which involves a complex system of engineering.

Research on the efficiency of green technology innovation in existing literatures can be summarized in the following three points: (1) the identification of factors and bottlenecks of green technology innovation efficiency; (2) the design of a green technology innovation efficiency evaluation index system; and (3) the construction of green technology innovation efficiency measurement methods.

At present, the analysis of the factors affecting the efficiency of green technology innovation is mainly concentrated on macro-policy and industrial enterprise. In terms of macroeconomic policies, scholars’ research mainly reveals the importance of government support [17,18], policy portfolio [19], and environmental rules [9] for green technology innovation. For industrial enterprises themselves, they should focus on corporate governance [13], green investment [20,21], social reciprocity [6], etc. In order to make an assessment of green technology innovation efficiency, a reasonable index system should be established. The existing research on green technology innovation efficiency evaluation index systems is mainly focused on integrating environmental factors with other factors related to technological innovation, and constructing new green performance evaluation index systems. From the four aspects (i.e., management innovation, process innovation, product innovation, technological innovation), Tseng et al. [22] discussed green innovation and then constructed the enterprise green technology innovation efficiency evaluation system including 22 indicators. Some of the indicators are listed as follows: investment in green equipment and technology; implementation of a comprehensive materials saving plan; a supervision system and technology transfer; advanced green production technology; and management of documentation and information. Luo and Liang [23] constructed a green technology innovation efficiency evaluation system for regional industrial enterprises in China from the perspective of green technology innovation input, intermediate output, expected output, and undesired output. Although these indicators are comprehensive, their inherent logical structure is unreasonable, and it is difficult to evaluate the efficiency of green technology innovation effectively. According to the innovation value chain [24], green technology innovation generally includes two stages: green technology R&D and green technology transformation. However, the existing research mostly regards it as a single-stage or whole-stage green technology innovation efficiency to measure, and few literatures divide it into two-stage efficiency for further study [25].

In order to scientifically and quantitatively evaluate green technology innovation, many methods are constructed to measure green technology innovation efficiency. According to the concept and definition of green technology innovation, there are three main types of measurement methods for green technology innovation efficiency: (1) Based on the achievements of green technology innovation, a single indicator method such as green technology patents are used to measure changes in green technology innovation efficiency [26]. However, due to the wide range of green technology innovation activities, it is difficult to use a single indicator to fully reflect the efficiency level of green technology innovation. (2) Based on the principal component analysis method, the green technology innovation efficiency of regions, industries, and companies is measured [23,27]. Compared with the single indicator measurement method, this method can fully reflect the efficiency of green technology innovation. However, green technology innovation is a dynamic process. This kind of method cannot reflect the internal operation mechanism of enterprise technology innovation activities, and it may fail to reflect the characteristics of different stages. (3) From the perspective of input–output, some non-parametric methods and parametric methods are used to measure green technology innovation efficiency. These non-parametric methods mainly include DEA [28,29,30], while parametric methods mainly include stochastic frontier analysis (SFA) [31,32,33]. Considering the stage characteristics of green technology innovation, it is difficult to effectively measure and evaluate the efficiency of green technology innovation using traditional parametric and non-parametric methods.

According to the above analysis and discussion, there still exists the following shortcomings: (1) The staged nature of green technology innovation efficiency and the relevance of shared input should be fully considered. When research evaluates the efficiency of green technology innovation, most of the indicators are regarded as a single stage or a whole stage, rather than two stages [25]. In addition, the original technology innovation investment will not only promote the intermediate output such as patents, but also affect the second stage of green technology innovation output, and the technology innovation investment will be shared in a certain proportion in two stages [34]. There is always a time lag effect in the input-to-output of technology, which means that the investment in technology usually takes some time to produce results. Therefore, it is necessary to consider the shared input factor and time lag in the two-stage DEA. (2) The green technology innovation index system should be improved. At present, the green technology innovation efficiency index system only considers industrial energy consumption and parts of the “three wastes” pollutants as environmental outputs [20,25], but ignores the environmental impact of carbon emissions. Therefore, industrial carbon emissions, all of the “three wastes” pollutants, digestion and absorption costs, and other factors should be fully considered in the process of constructing a green technology innovation index system. In order to overcome these shortcomings, we constructed a novel evaluation index system with industrial carbon emissions per unit of GDP and “three wastes” pollutants, using a two-stage network DEA model. we used all of these indictors to measure regional industrial enterprises’ green innovation efficiency and analyzed their differences.

## 3. An Industrial Enterprises’ Green Technology Innovation Efficiency Evaluation Index System

According to innovation value chain theory [24], industrial enterprises’ green technology innovation activities can be divided into two stages of green technology: R&D and green product transformation. The green technology R&D stage is a process in which enterprises use innovative resources to realize intermediate output such as patents, and mainly includes research, development, technology introduction, and absorption [35]. The green technology transformation stage is the process of transforming scientific and technological achievements into economic benefits and environmental optimization based on enterprise technology R&D, including production, technology industrialization, greening, and marketing.

The funding and manpower input in the green technology R&D stage not only has a direct impact on intermediate output such as patents, but also promotes the green economy output during the green achievement R&D stage. Besides, technical digestion and absorption also have a significant impact on the two-stage output. Therefore, the green technology innovation input factors such as technology R&D expenditure, R&D personnel full-time equivalent, and technology digestion and absorption will be shared in a certain proportion in two stages. Figure 1 shows a two-stage green innovation activity for industrial enterprises based on shared input linkages. The efficiency of green technology R&D is the ratio of intermediate output (technical output) to a certain proportion of green technology innovation investment. It is used to measure the ability of companies to convert a certain percentage of green technology innovation inputs into technological output [36,37,38,39,40]. The efficiency of green technology achievement transformation is the ratio of green technology innovation output to the sum of the intermediate output and the remaining part of the technological innovation input. It reflects industrial enterprises’ ability to transform the remaining part of the innovation investment and intermediate output into a green economy.

Based on the innovative value chain theory [24], in the green technology R&D stage, industrial enterprises need to have certain innovation inputs in order to obtain intermediate output. Finally, they can realize a green economy. Similarly, we can apply this model to analyze industrial enterprises’ green technology innovation input, intermediate output, and final output. Green technology innovation investment. The technological innovation investment is mainly reflected in the two aspects of manpower and capital. Research and development investment is mainly measured by R&D expenses and R&D workers. It is an important form of innovation resources. Relevant research shows that R&D investment is closely related to green innovation efficiency [36]. Considering the impact of each indicator on the intermediate output and the final output, this paper finally selects four indicators from the four perspectives: R&D workers, R&D funds, digestion and absorption fees of technology introduction, and new product development expenditures. In addition, the four indicators are described as follows: R&D personnel full-time equivalent (refers to the sum of the workload of full-time R&D workers and the workload of part-time workers converted according to actual working hours), the stock of R&D expenditure, the stock of imported digestion and absorption expenses (refers to the work carried out on the mastery, application, and reproduction of imported technologies, as well as innovations based on them), and the stock of new product development funds are selected as indexes for green technology innovation input of industrial enterprises. Where, the stock refers to the balance of products, goods, reserves, assets, and liabilities that were produced and accumulated in the past at a specified time.Intermediate output. The intermediate output of technological innovation in industrial enterprises is generally reflected by patents and new product development projects [37,38]. Some studies have shown that patents play a role in the development and diffusion of green technologies [39]. The patents mainly include indicators such as the number of patent applications and patents granted (especially the number of valid invention patents). The number of patent applications and valid invention patents are important indicators for measuring the technological innovation and technological output of enterprises. The number of new product development projects is often used to measure the ability of industrial companies to convert R&D investment into technology that can be exploited. Therefore, this study selects the number of patent applications, the number of valid invention patents (refers to a patent that is still in a valid state after the patent application is authorized), and the number of new product development projects as the intermediate output indicators of industrial enterprises’ technological innovation.Final output. The output effect of industrial enterprises’ green technology innovation is mainly measured by economy and environment [20]. We usually use the sales revenue of new products and the income from the main business of industrial enterprises to measure economic output. The sales revenue of new products reflects the achievements of enterprises in the field of product innovation. But the increase of output (or income) brought by some small inventions and process improvements cannot be reflected by the new products’ sales revenue. So, it also needs to include the main business income indicators [41]. Because industrial energy consumption and some “three wastes” pollutants and carbon emissions have direct and important impacts on the environment, we selected them as environmental output indexes in the efficiency research framework.

Based on the above analysis and discussion, with references to the relevant literature [18,26,34,37], in this paper, R&D personnel full-time equivalent, the stock of R&D expenditure, the stock of imported digestion and absorption expenses, and the stock of new product development funds were chosen as the indexes of industrial enterprises’ technological innovation input, and the number of patent applications, the number of valid invention patents, and the number of new product development projects were taken as intermediate output indexes, while new product sales revenue and main business income were determined as economic output indexes, and industrial carbon emissions of per unit industrial GDP and industrial “three wastes” pollutants of per unit industrial GDP were considered as final environmental output indexes. This indicator system completely describes the whole process of industrial enterprise green technology innovation (from the previous capital and manpower input to the medium-term patent output and the later green product output) and fully considers economic and environmental factors. Thus, we have established a two-stage efficiency evaluation index system for industrial enterprises’ green technology innovation, as shown in Table 1.

## 4. A Two-Stage Network DEA Model with Shared Input

Enterprise green technology innovation is a multi-stage complex system with intermediate input–output elements and subsystems. Traditional DEA models take the enterprise green technology innovation efficiency system as a “black box” and do not consider the reinvestment and sharing of the intermediate products. Thus, it is impossible to understand the impact of each efficiency sub-phase of its internal operation process on the overall system efficiency, and the source of efficiency loss cannot be determined [42]. The manpower and capital investment of green technology innovation not only affects the intermediate output, but also contributes to the enterprise green technology innovation output. Green technology innovation investment is shared in a certain proportion in two stages. The two-stage DEA model with shared input decomposes the whole process of the decision-making unit into several sub-processes or stages. Each stage is distinguished by its own input and output process, and all stages are related through intermediate elements [41]. Due to the superiority of the two-stage network DEA model with shared input, we exploited it to open the “black box” of the two-stage green technology innovation efficiency of industrial enterprises.

Assume that there exist n decision module units (DMUs), and each DMU includes m green technology innovation investments, q intermediate outputs, and s ultimate green technology innovation outputs. Let $DMU_{j},x_{ij},z_{pj},y_{rj}$ stand for the j(j=1,2,…,n) DMU, and i(i=1,2,…,m) stand for innovation investment, p(p=1,2,…,q) stand for intermediate output, and r(r=1,2,…,s) stand for ultimate green technology innovation output of the DMU j, respectively. In the process of green technology innovation, the initial green technology innovation investments are not completely consumed in the green R&D stage, they are often distributed in two- stage efficiency within certain proportions. Assume that $\alpha_{i}X_{ij}$ and $(1-\alpha_{i})X_{ij}$ express the discretionary inputs of the green technology R&D and technological achievements transformation, respectively, and $v_{i}^{1},v_{i}^{2}(i=1,2,\ldots ,m),u_{r}(r=1,2,\ldots ,s)$ indicates the weights of the green technology innovation inputs $\alpha_{i}X_{ij}$ and $(1-\alpha_{i})X_{ij}$, and the final green technology innovation output $y_{rj}$, respectively. The intermediate products are both the output of the green technology R&D stage and the input of the technological achievements transformation stage. In this paper, we used $w_{p}^{1},w_{p}^{2}(p=1,2,\ldots ,q)$ to indicate the weights of the intermediate output in the two stages. $DMU_{j}(j=1,2,\ldots ,n)$ inputs and outputs in the green technology R&D phase were $\sum_{i=1}^{m}v_{i}^{1}\alpha_{i}X_{ij}$ and $\sum_{p=1}^{q}w_{p}^{1}Z_{pj}$. The inputs and outputs in the transformation of technological achievements were $\sum_{i=1}^{m}v_{i}^{1}(1-\alpha_{i})X_{ij}+\sum_{p=1}^{q}w_{p}^{2}Z_{pj}$ and $\sum_{r=1}^{s}u_{r}^{1}Y_{rj}$.

According to linear programming theory and the input and output principle of green technology innovation, under the background of variable scale, we can obtain the green R&D efficiency $E_{k}^{1}$ of the kth decision-making unit ($DMU_{k}$) as follows:(1)$$E_{k}^{1}=(\max\sum_{p=1}^{q}w_{p}^{1}Z_{pk}-\mu_{1})/\sum_{i=1}^{m}v_{i}^{1}\alpha_{i}X_{ik}$$

Let $t=1/\sum_{i=1}^{m}v_{i}^{1}\alpha_{i}X_{ik}$, $V_{i}^{1}\alpha_{i}=\pi_{i}^{1}$, $V_{i}^{2}\alpha_{i}=\pi_{i}^{2}$, by the Charnes–Cooper transform, we can convert formula (1) into a linear form. By doing this, the optimal value of the green technology R&D efficiency of the $DMU_{k}$ in the variable scale returns can be obtained as follows:(2)$$\begin{array}{l} E_{k}^{1}=\max\sum_{p=1}^{q}w_{p}^{2}Z_{pk}-\mu_{A}\\ s.t\left\{ \begin{array}{l} \sum_{i=1}^{m}\pi_{i}^{1}X_{ik}=1\\ \sum_{i=1}^{m}\pi_{i}^{1}X_{ij}-(\sum_{p=1}^{q}w_{p}^{1}Z_{pj}-\mu_{A})\geq 0,j=1,2,\ldots ,n\\ \sum_{i=1}^{m}V_{i}^{2}X_{ij}-\sum_{i=1}^{m}\pi_{i}^{2}X_{ij}+\sum_{p=1}^{q}w_{p}^{2}Z_{pj}-(\sum_{r=1}^{s}U_{r}Y_{rj}-\mu_{B})\geq 0,j=1,2,\ldots ,n\\ v_{i}^{2}\geq \pi_{i}^{2}\geq \varepsilon ;U_{r},\pi_{i}^{1}W_{P}^{1}W_{P}^{2}\geq \varepsilon ,i=1,2,\ldots ,m \end{array}\right. \end{array}$$
where, $V_{i}^{1}=tv_{i}^{1}$, $V_{i}^{2}=tv_{i}^{2}$, $W_{p}^{1}=tw_{p}^{1}$, $W_{p}^{2}=tw_{p}^{2}$, $U_{r}=tu_{2}$, $\mu_{A}=t\mu_{1}$, $\mu_{B}=t\mu_{2}$.

For formula (2), the green R&D efficiency of the decision-oriented evaluation unit based on input orientation can be calculated. In the same way, the optimal results of the transformation efficiency of $DMU_{k}$ can be determined as follows:(3)$$\begin{array}{l} E_{k}^{2}=\max\sum_{r=1}^{s}U_{r}Y_{rk}-\mu_{B}\\ s.t\left\{ \begin{array}{l} \sum_{i=1}^{m}V_{i}^{2}X_{ik}-\sum_{i=1}^{m}\pi_{i}^{2}X_{ik}+\sum_{p=1}^{q}W_{p}^{2}Z_{pk}=1\\ \sum_{i=1}^{m}\pi_{i}^{1}X_{ij}-(\sum_{p=1}^{q}w_{p}^{1}Z_{pj}-\mu_{A})\geq 0,j=1,2,\ldots ,n\\ \sum_{i=1}^{m}V_{i}^{2}X_{ij}-\sum_{i=1}^{m}\pi_{i}^{2}X_{ij}+\sum_{p=1}^{q}w_{p}^{2}Z_{pj}-(\sum_{r=1}^{s}U_{r}Y_{rj}-\mu_{B})\geq 0,j=1,2,\ldots ,n\\ v_{i}^{2}\geq \pi_{i}^{2}\geq \varepsilon ;U_{r},\pi_{i}^{1}W_{P}^{1}W_{P}^{2}\geq \varepsilon ,i=1,2,\ldots ,m \end{array}\right. \end{array}$$
where, $V_{i}^{1}=t^{\prime }v_{i}^{1}$, $V_{i}^{2}=t^{\prime }v_{i}^{2}$, $W_{p}^{1}=t^{\prime }w_{p}^{1}$, $W_{p}^{2}=t^{\prime }w_{p}^{2}$, $U_{r}=t^{\prime }u_{2}$, $\mu_{A}=t^{\prime }\mu_{1}$, $\mu_{B}=t^{\prime }\mu_{2}$, $t^{\prime }=1/(\sum_{i=1}^{m}V_{i}^{2}X_{ik}-\sum_{i=1}^{m}\pi_{i}^{2}X_{ik}+\sum_{p=1}^{q}W_{p}^{2}Z_{pk})$.

According to the calculation results of the green technology R&D efficiency $E_{k}^{1}$, the technological achievement transformation efficiency $E_{k}^{2}$, and the green technology innovation comprehensive efficiency $E_{k}(E_{k}=E_{k}^{1}\times E_{k}^{2})$ of industrial enterprises, we discover the specific links regarding the loss of green technology innovation efficiency for regional industrial enterprises in China, so that it can provide a scientific basis for the targeted formulation of green technology innovation policies.

## 5. Empirical Analysis

### 5.1. Object Selection and Data Processing

In this paper, we select industrial enterprises from 30 provinces in mainland China as research objects (as Tibet, Hong Kong, Macao, and Taiwan are not included, the data is not complete). According to the China Science and Technology Statistical Yearbook [37], the China Industrial Economics Statistical Yearbook [37], the China Statistical Yearbook [37], the China Environmental Statistics Yearbook [37], and the China Energy Statistics Yearbook in 2010–2017 [37], we collected related data on green innovation efficiency evaluation indexes and standardized them using a two-stage network DEA model with shared input to measure the efficiency of China’s industrial enterprises’ green technology innovation, and then analyzed regional differences.

Green technology innovation investment should be measured from the perspective of human resources and capital investment. In terms of human resources, the R&D personnel full-time equivalent was selected as the composition of innovation input in the indicator system. In terms of capital investment, the internal R&D expenditure, the expenditure of technology introduction, digestion and absorption, and the expenditure of new product development were selected as the composition of innovation input in the indicator system. Among them, the expenditure of technology introduction and digestion was equal to the sum of the expenditure for introducing technical funds, the expenditure for digestion and absorption, the expenditure for purchasing domestic technology, and the expenditure for technological transformation. In addition, R&D expenditure has a time lag effect, that is, the current R&D expenditure will not only affect the current industrial green innovation output, but also the industrial green innovation output in the future. However, there is no strict standard for choosing how long the lag period will be. Considering that the stock index method can reflect the lagging effect of R&D investment to a certain extent [43,44], this study used the perpetual inventory method to characterize and measure the stock index of R&D capital investment. We select the number of patent applications, valid invention patents, and new product development projects to represent the intermediate output of industrial enterprises’ green technology innovation. The output of green technology innovation was mainly measured from the perspective of economic output and environmental benefits. We selected new product sales revenue and main business income to indicate economic output. Here, the 2009 constant price industrial product ex-factory price index was used for deflation. The per unit industrial carbon emissions of industrial GDP and the per unit industrial “three wastes” emissions of industrial GDP were selected to represent environmental output. Specifically, we select 15 kinds of industrial carbon emissions as energy benchmarks, which are listed as follows: raw coal, clean coal, other coal washing, coke, coke oven gas, other gas, crude oil, gasoline, kerosene, diesel, fuel oil, liquefied petroleum gas, natural gas, heat, and electricity. The formula is as follows:(4)$$CO_{2}=\sum_{i=1}^{15}EN_{i}\times NCV_{i}\times CEF_{i}\times COF_{i}\times (44/12)$$
where, $EN_{i}$ stands for the energy consumption of 15 kinds of industrial coal, such as raw coal, clean coal, coke, and electricity. $NCV_{i},CEF_{i}$ express the average low calorific value of energy, the carbon emission factor given in the IPCC (Intergovernmental Panel on Climate Change) National Greenhouse Gas Emissions Inventory 2006, respectively. $COF_{i}$ stands for a carbon oxidation factor and had a value of 1, while 44/12 is the gasification coefficient of $CO_{2}$. Then, dividing the industrial carbon emissions by the deflated industrial GDP, the per unit carbon emissions of industrial GDP were obtained. Based on the negative normalized method, we can convert the obtained data to values between (1,100). In addition, we selected five indexes of waste gas of industrial GDP (after reduction), industrial waste water, industrial solid waste, industrial $SO_{2}$, and industrial smoke dust, and exploited the entropy method to calculate the industrial “three wastes” pollutant emissions of per unit industrial GDP. Because the larger three waste pollutants of per unit GDP, the lower environmental output level, and the measurement units of the indicators are not uniform, they need to be standardized. Through the entropy method with negative normalization, the related data can be converted into (1,100).

In addition, in order to fully demonstrate the scale of industrial enterprises, the number of industrial enterprises in various provinces in China during the observation period are shown in Table 2.

### 5.2. Measurement and Regional Difference Analysis of Industrial Enterprises’ Green Innovation Efficiency

#### 5.2.1. Measurement of Regional Industrial Enterprises’ Green Innovation Efficiency

From the perspective of two-stage green technology innovation with sharing input, by considering industrial carbon emissions, all “three wastes” pollutants, digestion and absorption costs, and other factors, we exploited the two-stage network DEA model with variable scale returns and Python software programming to calculate the efficiency of the green technology innovation R&D stage, transformation stage, and the comprehensive green technology innovation efficiency of industrial enterprises in 30 provinces in China from 2009 to 2016, and the specific data are shown in Table 3, Table 4 and Table 5.

### 5.2.2. Regional Differences Analysis of Industrial Enterprises’ Green Technology Innovation Efficiency

According to the green technology innovation efficiency of industrial enterprises across the country and all regions, the overall R&D, transformation efficiency, and comprehensive efficiency value of green technology innovation in China and the four regions can be calculated. As it can be seen in Figure 2, from 2009–2016, China’s industrial enterprises’ green technology innovation R&D efficiency was 0.855, achievement transformation efficiency value was 0.926, and the average of the overall efficiency was 0.791.

It shows that there was still some room for further improvement in industrial enterprises’ efficiency for two-stage green innovation. Efficiency loss in the green R&D stage reached 14.5%, which was the main reason for the low overall efficiency for green technology innovation in Chinese enterprises. The comparison of green technology innovation efficiencies in the four major economic regions during the R&D stage is shown below. The western region (average efficiency was 0.906) and the eastern region (average efficiency was 0.898) had relatively high green R&D efficiency, followed by the central region (average efficiency was 0.799). In the north-east region (average efficiency was 0.636), the efficiency of enterprises was obviously low, and the efficiency loss gap was as high as 36.4%. In the stage of green technology achievement transformation, the efficiency ranking was as follows: the eastern region (average efficiency was 0.977), the central region (average efficiency was 0.966), the western region (average efficiency was 0.889), and the north-east region (average efficiency was 0.815). Overall, the efficiency of China’s four major economic zones is at a relatively high level. However, the efficiency of the western and north-east regions still need some improvement. Since this study incorporates the industrial carbon emissions of unit GDP and “three wastes” pollutants into the performance evaluation index system and adopts the two-stage network DEA model of shared input, some problems that were ignored in previous studies have been taken into account. The analysis results concluded above are not completely consistent with previous research results: “Efficiencies in the eastern, central and western regions are declining”, and some further studies have reached similar conclusions [34,36]. Although there are a small number of industrial enterprises in the western region, they have more national key laboratories, scientific research institutions, and national key support policies and funds, while the degree of environmental damage is small [34,37]. The eastern region is a developed region with high levels of science and technology. The introduction of foreign-funded enterprises has also brought about certain technological demonstrations and spillover effects, and the innovation environment and marketization process has continuously improved. Therefore, enterprises in these regions have achieved intermediate outputs such as patents and new product development projects under the established R&D funds and human input, promoting their green R&D efficiency to a higher level. The central region and the north-east region are relatively lacking in government support and development opportunities. Their innovative environment, technical management level and intellectual property protection are not as good as those in the eastern and western regions. In addition, the central and western regions’ patent outputs and other output indicators are seriously inadequate, and there is still space for improvement. The main problem for enterprise technology innovation development in the western region is that although the green technology innovation rate is at a relatively high level for the two stages, its input and output volume is small, and the technological innovation structure is relatively simple.

Figure 3, Figure 4 and Figure 5 show the trends in green technology R&D efficiency, achievement transformation efficiency, and comprehensive efficiency change across the China and the four main regions. It can be seen from the figures that during the period from 2009 to 2016, the indicators of the four regions showed a decreasing trend in green technology innovation during R&D stage. The R&D efficiency of the western region and the eastern region was at a relatively high level—higher than the national level—while the efficiency of the central region and the north-east region was at a low level. The central region was higher than the national average, and the north-east region was obviously lower than the national average.

The analysis of green technology innovation efficiency in the transformation stage of the four major economic zones in the country is as follows. The eastern region, western region, and central region showed increasing trends, while the north-east region showed a decreasing trend. The eastern and central regions were generally higher than the national average, and the western and north-east regions were below the national average. The main reason for this is that the geographical advantages of the western region and the north-east region are not obvious, and the development of the green technology market is still insufficient, with a small volume and single demand. From the perspective of the comprehensive efficiency of green technology innovation, the overall efficiency of the four regions was generally decreasing in the development stage and the transformation stage. The eastern region has rich experience in the transformation of scientific and technological achievements, infrastructure, and environmental protection. Furthermore, the eastern region was always at a relatively high level of efficiency and led the country. However, as a traditional heavy industrial base, the north-east region was obviously underpowered in terms of technological transformation and green innovation.

### 5.2.3. Analysis of the Difference of Green Industrial Enterprises in Different Provinces

#### The Overall Difference of Industrial Enterprises’ Green Technology Innovation Efficiency

In order to comprehensively analyze the green technology innovation efficiency of industrial enterprises in different Chinese provinces and measure the difference in regional efficiency, we carry out cluster analysis on the average comprehensive efficiency of all provinces in China. Due to the limitations of complexity and uncertainty in reality, clustering analysis of green technology innovation efficiency is a typical panel data clustering problem. Therefore, we use the grey relational clustering method [45] for efficiency analysis, and the results are shown in Figure 6.

As can be seen from Figure 6, according to the comprehensive efficiency of industrial enterprises’ green technology innovation, 30 provinces in China were divided into four categories: high, good, medium, and poor. The provinces with “high efficiency” levels mainly include: Xinjiang, Inner Mongolia, Qinghai, Ningxia, Shanxi, Beijing, Tianjin, Sichuan, Chongqing, Jiangsu, Anhui, Shanghai, Zhejiang, Guangdong, Yunnan, and Hainan. The provinces with “good efficiency” levels mainly include: Hebei, Gansu, Shandong, Hubei, Jiangxi, and Guangxi. The provinces with “medium efficiency” levels mainly include: Jilin, Liaoning, Henan, Guizhou, Hunan, and Fujian. The provinces with “poor efficiency” levels mainly includes: Heilongjiang and Shaanxi. It can be seen from the clustering results that the efficiency levels of green technology innovation in China’s provinces was not completely in line with the previous research results that stated, “the eastern region has the highest efficiency, while the central and western regions have significantly lower efficiency” [34,36]. The provinces in the western region were obviously better than those in the central region and the north-east region, and there were also some provinces with low efficiency in the eastern region, such as Fujian. The probable reason is that previous studies only used single-stage DEA for analysis, and only considered part of the “three wastes” and industrial energy consumption indicators. In this paper, industrial carbon emissions and all of the “three wastes” pollutants were included in the efficiency research framework, and the investment sharing correlation between the two stages of green technology innovation was considered. It was found that in the green R&D stage, the industrial enterprises in the eastern and western regions were relatively more efficient, while the central regions and the north-east regions had greater efficiency losses. In the stage of green achievement transformation, the efficiency of enterprises in the eastern and central regions was at relatively high levels. The north-east region was obviously behind the national level, but there is room for further efficiency improvement.

#### Stage Difference Analysis of Industrial Enterprises Geen Technology Innovation Efficiency

Next, we analyzed the spatiotemporal differences in the efficiency of green technology innovation in different provinces. From 2009 to 2016, the change trend charts of R&D efficiency, achievement transformation efficiency, and comprehensive efficiency of green technology innovation in China’s 30 provinces were drawn, as shown in Figure 7, Figure 8 and Figure 9.

As can be seen from Figure 7, the R&D efficiency of Qinghai, Guangdong, Ningxia, Sichuan, Yunnan, Hainan, Beijing, Shanghai, Anhui, Zhejiang, and other provinces has always been above 0.900, and the R&D efficiency values were relatively high. These provinces mainly come from the eastern and western regions. For Zhejiang and Guangdong, they have achieved good results in scale and R&D efficiency, which means they are in a stable development trend. For enterprises in Qinghai, Ningxia, and Hainan, although the R&D output of patents was less than that of enterprises in eastern coastal areas, the efficiency of green technology innovation input resource utilization was high and the management was appropriate. The efficiency level of Gansu, Tianjin, Jiangsu, Chongqing, and Guizhou provinces fluctuated greatly. The R&D efficiency of Shandong, Shanxi, Shaanxi, Guangxi, Hubei, Hunan, Hebei, and Guangxi provinces shows a decreasing trend during the observation period, which requires the attention of relevant departments to explore the deep causes. The R&D efficiencies of Liaoning, Jilin, Heilongjiang, and Henan provinces are always at a low level during the observation period, and the efficiency value is less than 0.700. Great improvements should be made in technology innovation R&D and patents output, as these provinces mainly come from the central and north-east regions.

According to Figure 8, during the observation period, Inner Mongolia, Jilin, Beijing, Tianjin, Shanghai, Zhejiang, Shandong, Qinghai, Hebei, Henan, Chongqing, Sichuan, Guangdong, Hainan, Jiangsu, and other provinces have relatively high transformation efficiency. In addition, provinces with high transformation efficiency are more than provinces with high R&D efficiency. From 2009 to 2012, the efficiency of achievement transformation in Gansu, Guangxi, and Guizhou provinces was low. During the observation period, the efficiency of Ningxia, Fujian, Guangxi, and other provinces fluctuated greatly. The efficiency of achievement transformation in Heilongjiang and Shaanxi was always below 0.500, and the transformation ability was seriously insufficient. Great improvements are needed in economic output and environmental protection. In general, the transformation efficiency of green technology achievements in each province is better than the R&D efficiency, which is consistent with the previous analysis.

From Figure 9, during the observation period, the comprehensive efficiency of green innovation in Beijing, Tianjin, Jiangsu, Shanghai, Qinghai, Guangdong, Hainan, Zhejiang, Anhui, Ningxia, and other provinces was greater than 0.800, and the comprehensive efficiency value was relatively high. In combination with Figure 7 and Figure 8, the R&D efficiency and the transformation efficiency value of achievements were relatively high. These provinces were the benchmarks for green technological innovation in China. It is of great importance to study and summarize the experiences of R&D transformation for technological innovation for guiding the development of green technological innovation in other provinces. Heilongjiang, Shaanxi, Hebei, Henan, Hunan, Guangxi, Liaoning, Jilin, Fujian, Jiangxi, and other provinces had low comprehensive efficiency values. According to Figure 7 and Figure 8, this was mainly due to the low efficiency of green R&D. Therefore, these provinces need to strengthen their R&D capacity in the future, introduce high-tech enterprises, universities, and scientific research institutions, and improve the number of patent applications in R&D output.

#### Regional Difference Analysis of Industrial Enterprises’ Green Technology Innovation Efficiency under Different Combination Models

According to the above analysis, we know that green technology innovation in R&D efficiency and results transformation efficiency affect the ultimate comprehensive efficiency. Although some provinces have higher achievement transformation efficiency, their R&D efficiencies are at a lower level, which will resulting in lower comprehensive efficiency. Therefore, it is necessary to further analyze the relationship between R&D of green technology and transformation of green achievements of industrial enterprises in China’s provinces, and to comprehensively analyze the differences from these two dimensions. Taking the two-stage average efficiency (0.855 and 0.925) of all provinces as the demarcation point, the green technology innovation of each province was divided into four combination modes: high green R&D–high achievement transformation; low green R&D–high achievement transformation; low green R&D–low achievement transformation; and high green R&D–low achievement transformation, as detailed in Figure 10.

High green R&D and high achievement transformation. This category includes Inner Mongolia, Tianjin, Xinjiang, Shanghai, Shanxi, Jiangsu, Qinghai, Hainan, Guangdong, Sichuan, Anhui, Yunnan, Zhejiang, and accounts for about 43.3% (the proportion of provinces in this category to 30 provinces), mainly from the eastern and western regions. Industrial enterprises in these regions show high R&D efficiency and achievement transformation efficiency during the conversion stage, which belongs to an efficient and intensive green technology innovation and development mode. While maintaining their own advantages, these regions should vigorously develop the technological innovation of public goods from relatively weak links, and follow-up with international leading green technology and management experience to occupy the commanding heights of green technology innovation. In addition, although the efficiency of Inner Mongolia, Xinjiang, Qinghai, Hainan, and Yunnan provinces were at a high level in the two stages, their economies are small and their technological structures are single. In the future, they need to broaden their areas of technological innovation according to local conditions and strengthen links with the eastern region to open up new markets.

Low green R&D and high achievement transformation. This category included Henan, Jilin, Liaoning, Hubei, Jiangxi, Shandong, Hebei, and accounted for about 23.3% (the proportion of provinces in this category to 30 provinces), mainly from the central and north-east regions. These regions had high efficiency in the stage of achievement transformation, but low efficiency in the stage of green R&D, which restricted improvements in green technology innovation abilities. Such areas should start from the green research and development stage, strengthen the depth and breadth of industry–university–research cooperation, attract high-tech talents, and improve patent outputs, while technology is introduced and digested.

Low green R&D and low achievement transformation. This category included Hubei, Guangxi, Gansu, Fujian, Shaanxi, Heilongjiang, and accounted for about 20% (the proportion of provinces in this category to 30 provinces), mainly from the central and western regions. In these areas, the efficiency of green R&D and the efficiency of achievement transformation were low, thus they need to make great improvements in both aspects of green R&D and achievement transformation, from low R&D-low conversion to high R&D-low conversion or low R&D-high conversion type, then to high R&D-high conversion type. If the scientific and technological economic foundation is good, they can try to transition directly from low R&D-low conversion type to high R&D-high conversion type. In addition, as can be seen from the lower left corner of Figure 10, Heilongjiang lags far behind other provinces in terms of green R&D efficiency and achievements conversion efficiency. During the observation period, there was no significant progress in its green technology innovation, thus more attention needs to be paid to its development issues.

High green R&D and low achievement transformation. This category includes Chongqing, Ningxia, Guizhou, and accounts for approximately 10% (the proportion of provinces in this category to 30 provinces), mainly from the western region. These provinces have higher efficiency in the green R&D stage, but lower efficiency in the transformation stage. Therefore, they need to start from expanding the market, pay attention to the construction of scientific and technological achievements transformation platform, create a good environment for green innovation, and establish a good awareness of cleaner production.

The above four modes of green technology innovation efficiency for each province was based on the mean value during the observation period. In order to clearly show the dynamic changes in the modes of each province, and to study the relationship between the efficiency of green R&D and the efficiency of achievement transformation in each province more carefully, the mode division of green technology innovation efficiency of the 30 provinces in China was carried out for each year, as shown in Figure 11. From Figure 11, we can see that Guangdong, Hainan, Zhejiang, Qinghai, and Beijing were in the high green R&D and high achievement transformation category during the observation period. This shows that the five provinces have maintained a high level of efficiency in the green R&D stage and the achievement transformation stage, and formed a stable and good development trend. In the future, they can catch up with world-class technology companies and occupy the commanding heights of technology while maintaining the efficiency of existing green technology innovation. Heilongjiang during the observation period was always in the low green R&D and low achievements transformation category. According to the above analysis, the achievements transformation stage efficiency of Heilongjiang province was obviously low. Moreover, in the long run, Heilongjiang still has not separated from the low green R&D and low achievement transformation mode. In the future, it should concentrate on green R&D or the transformation of achievements. On the one hand, it should seek to make breakthroughs for transform from high R&D-low conversion or low R&D-high conversion type to high R&D-high conversion type. For Shaanxi, Shanxi, Guizhou, Guangxi, Hunan, Chongqing, Gansu, Ningxia, Xinjiang, Yunnan, and other provinces, their green technology innovation mode is in a state of change during the observation period. These provinces are not in the stage of stable technological innovation, so they need to start looking at their weaknesses and develop their strengths. Henan, Hebei, Shandong, Hubei, and other provinces were mostly in the low green R&D and high achievement transformation category. These provinces need to pay attention to the improvement of green R&D efficiency, learn from the development experience of other green R&D efficient provinces, employ top international talent as technical consultants, maintain the advantages gained from the transformation stage, and strive to transform to a high R&D-high conversion type.

## 6. Conclusions and Suggestions

In this paper, from the perspective of a two-stage innovation value chain, by introducing the industrial carbon emissions per unit of GDP and the “three wastes” pollutants into the research framework of green technology innovation efficiency, we established a novel green innovation efficiency evaluation index system for industrial enterprises. Then we used a two-stage network DEA with shared input to measure the efficiency of regional enterprises’ green technology innovation and explored the efficiency of regional differences in industrial enterprises’ green technology R&D and green technology achievement transformation. The results show that: (1) The green innovation efficiency of Chinese industrial enterprises shows significant regional imbalances and differences. From 2009–2016, China’s industrial enterprises’ green technology innovation R&D efficiency was 0.855, achievement transformation efficiency was 0.926, and the average overall efficiency was 0.791. The efficiency of the two stages is not optimal, which leads to low comprehensive efficiency. There is still room for improvement in green technology R&D efficiency, which is also the main reason for the low comprehensive efficiency of green technology innovation. The efficiency of green technology innovation is not completely consistent with the previous research result. Specifically, the eastern and western regions were more efficient than the central region, and the efficiency of the north-east region was the lowest. (2) The provinces with low R&D-low achievement transformation accounted for 20% of the total, which mainly include Hubei, Guangxi, Gansu, Fujian, Shaanxi, and Heilongjiang in the central and western regions. In the green R&D stage and the green achievement transformation stage, 33% of the provinces (ie Henan, Jilin, Liaoning, Hubei, Jiangxi, Shandong, Hebei, Chongqing, Ningxia and Guizhou) have a certain degree of efficiency loss. (3) From 2009 to 2016, Guangdong, Hainan, Zhejiang, Qinghai, and Beijing always belonged to the high green R&D and high achievement transformation category. These five provinces maintained a high level of green R&D and achievement transformation, forming a stable and good development trend. During the observation period, Heilongjiang’s efficiency in the green R&D stage and efficiency of the green results transformation stage lagged far behind other provinces. It is very difficult for Heilongjiang to break away from the current status of low green R&D-low achievement transformation, and it may be in this state for a long time in the future. Some provinces such as Heilongjiang should be the focus of China’s green innovation policy.

According to the analysis and conclusions, some suggestions are provided as follows:From the perspective of enterprises, enterprises should rationally allocate people, information, capital, and other investment factors, pay attention to the sharing of green innovation resources in two stages, and strengthen the interaction between the two stages. Specifically, enterprises should focus on the management of green patent outputs, technology incubation, and especially the original green technology development activities, and create new green technology from 0 to 1. Regarding enterprises as the main body of green technology innovation is not to deny the important role of research institutes and universities in technological innovation, especially their important leading role in technological breakthroughs. Large companies need to further streamline and optimize their organizational structure, increase sensitivity to new markets and new products, and strengthen investment in green technology development and clean production equipment. The performance appraisal of state-owned enterprises can not only focus on the number of patents, but also pay attention to the quality of patents. The enterprises can reward departments or employees who have made green and cost-effective technologies. The good functioning of a system requires the relative parties to work together around a goal. In the future, enterprises in the eastern region must vigorously introduce innovation, digest and absorb innovation and carry out original innovation. Enterprises should work hard to construct modern enterprise systems and management reform, increase the endogenous power of enterprises, and pay attention to improve the management level of green technology innovation. By doing this, these enterprises can drive the central and western regions to shift to a high-quality and low-pollution economic growth mode. Central and western enterprises need to strengthen cooperation with eastern enterprises and implement pairing assistance. At the same time, central and western enterprises need to introduce high-tech talent, pay attention to the marketization and commercialization of technology, and improve the volume and efficiency of green R&D and achievements transformation.The government needs to raise green technology innovation to the national strategic level, build a linkage system for green technology innovation, and truly use green technology innovation as an important means to enhance comprehensive competitiveness. The government needs to develop a series of policies and incentives to encourage green technology innovation, such as strengthening support for innovation, improving innovative incentives, and accelerating the cultivation of emerging industries to accelerate the process of innovation and knowledge creation in high-tech industries.

(1) Strengthen the construction of green technology innovation culture. An innovative cultural environment is very important for scientific and technological innovation. It can not only stimulate the enthusiasm of employees, but also enable R&D workers to share common goals and achieve their goals. We should integrate innovative ideas into the daily work of science and technology workers and create a work environment that is innovative and inclusive. At the same time, we must follow the inherent process of innovation and establish a scientific and rational innovation evaluation mechanism. In addition, promoting institutional construction and management of innovation can enhance the enthusiasm of overall scientific and technological innovation, improve the output efficiency of scientific and technological achievements, and ensure the smooth progress of the scientific and technological innovation process.

(2) Increase investment in green technology innovation. It is necessary to continuously invest in science and technology innovation in high-tech industries, establish a stable growth mechanism, and increase support for technological innovation. For example, special funds should be set up to support evaluated, potential, and promising technologies, with a focus on supporting the development of key technologies, common technologies, and forward-looking technologies. Therefore, we should strengthen supervision over special funds for technological innovation and increase the output rate and conversion rate of scientific and technological achievements. At the same time, we should continue to expand financing channels and increase finance investment through various ways such as attracting venture capital and cooperating with R&D, and by attracting more social investment by setting up special funds such as emerging industry venture capital guidance funds.

(3) Improve the industry–university–research cooperation innovation system. It is necessary to speed-up the establishment of industry–university–research R&D mechanisms led by enterprises.

(4) Optimize green innovation policy to ensure stability of the innovation environment. Besides increasing innovation support and preferential efforts, we should establish a diversified investment and financing system and risk sharing mechanism, increase the implementation of scientific and technological innovation policies, and establish a good public atmosphere and social culture that supports and respects innovation. The government should formulate relevant laws and regulations on intellectual property protection, technology intermediary services, transformation of scientific and technological achievements, and construction of integrated innovation platforms. Furthermore, by improving the laws and regulations on technological innovation, we can provide a good policy environment for innovation activities.

However, this article also has certain limitations. Due to the limitation of data acquisition, the evaluation indicator system for green technology innovation efficiency constructed in this paper is not comprehensive and detailed enough. Some relevant important green innovation indicators were not included, and innovation needs to be improved. From a realistic perspective, the lag period for different green technology innovation efficiency evaluation indicators is generally different. Therefore, how to construct a two-stage network DEA model with different lag periods and shared input to measure the efficiency of green technology innovation is our future research direction.

## Figures and Tables

**Figure 1 ijerph-16-00940-f001:**
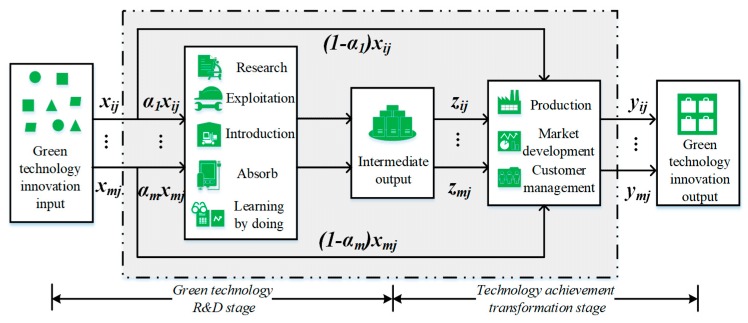
The two stages of industrial enterprises’ green innovation activities with shared input.

**Figure 2 ijerph-16-00940-f002:**
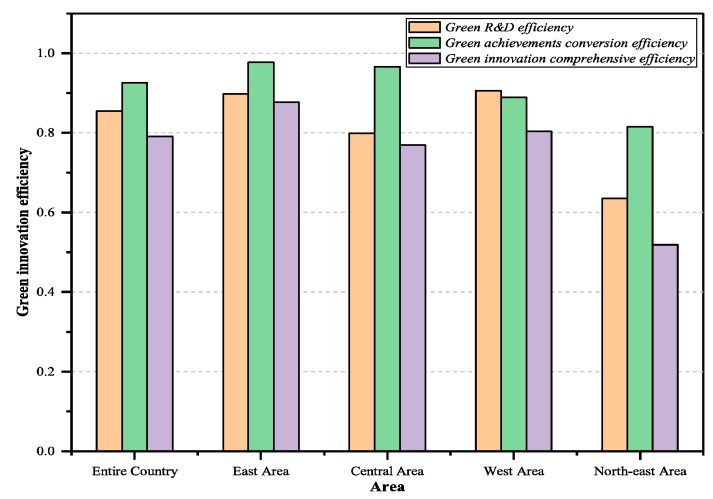
Green technology innovation efficiency of the country and sub-regions.

**Figure 3 ijerph-16-00940-f003:**
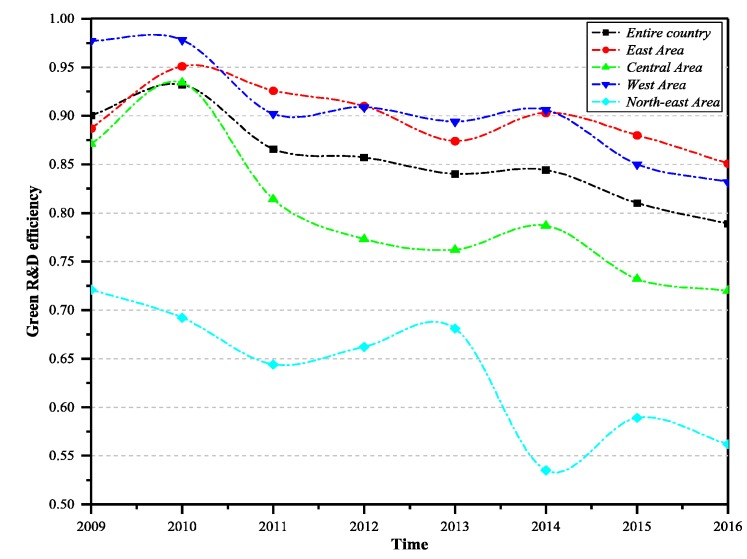
Green technology innovation R&D efficiency of the country and the four major regions (2009–2016).

**Figure 4 ijerph-16-00940-f004:**
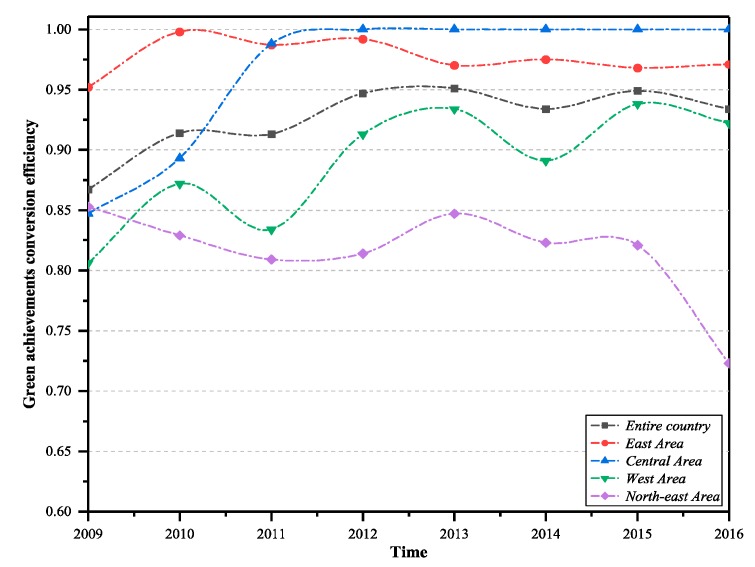
Green technology innovation achievements in transformation efficiency of the country and the four major regions (2009–2016).

**Figure 5 ijerph-16-00940-f005:**
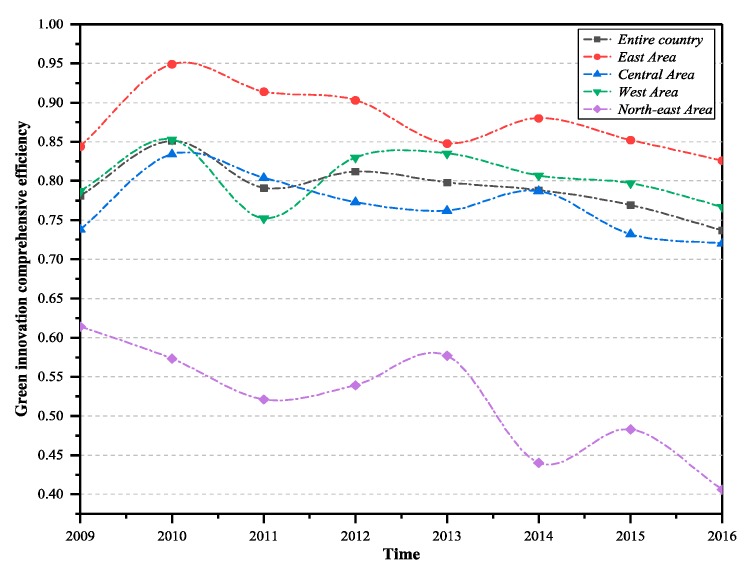
Green technology innovation comprehensive efficiency of the country and the four major regions (2009–2016).

**Figure 6 ijerph-16-00940-f006:**
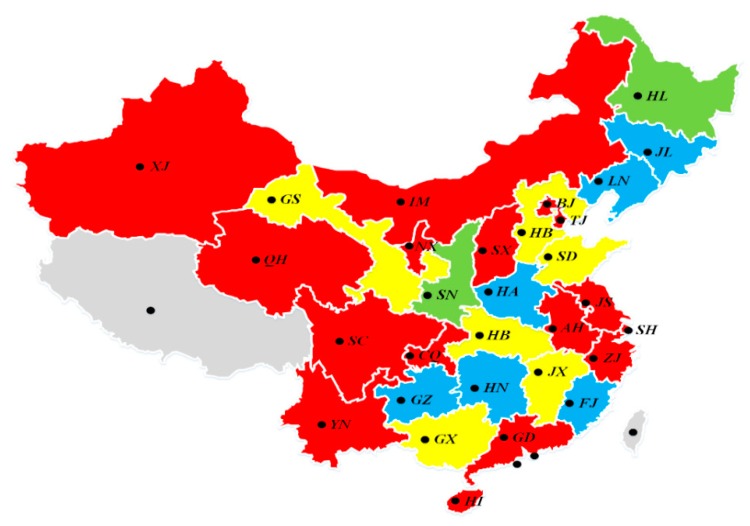
Clustering map of industrial enterprises’ green innovation comprehensive efficiency in China.

**Figure 7 ijerph-16-00940-f007:**
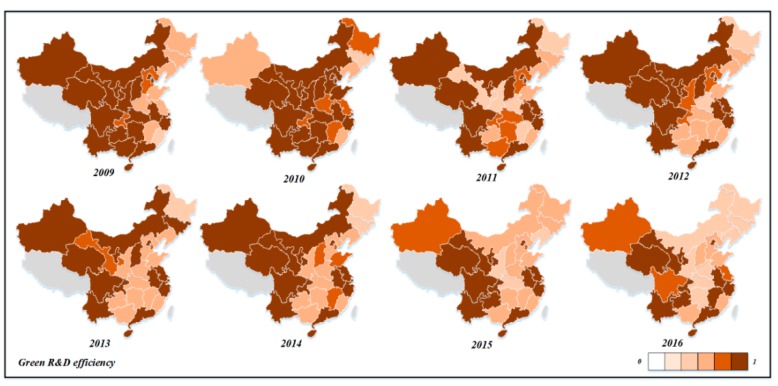
The evolution map of industrial enterprises’ green technology innovation in R&D efficiency in China (2009–2016).

**Figure 8 ijerph-16-00940-f008:**
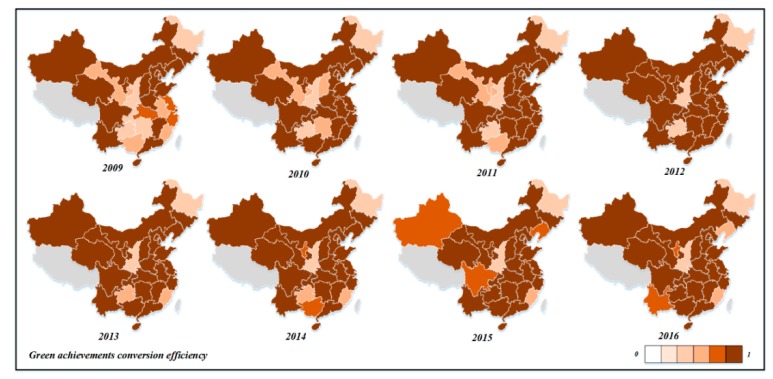
The evolution map of industrial enterprises’ green technology achievements in transformation efficiency in China (2009–2016).

**Figure 9 ijerph-16-00940-f009:**
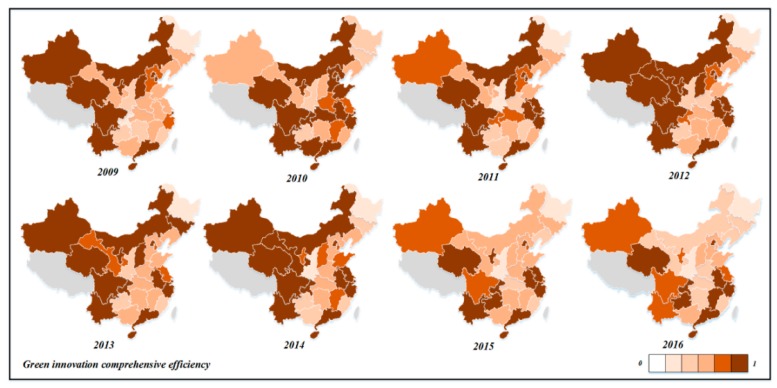
The evolution map of industrial enterprises’ green technology innovation comprehensive efficiency in China (2009–2016).

**Figure 10 ijerph-16-00940-f010:**
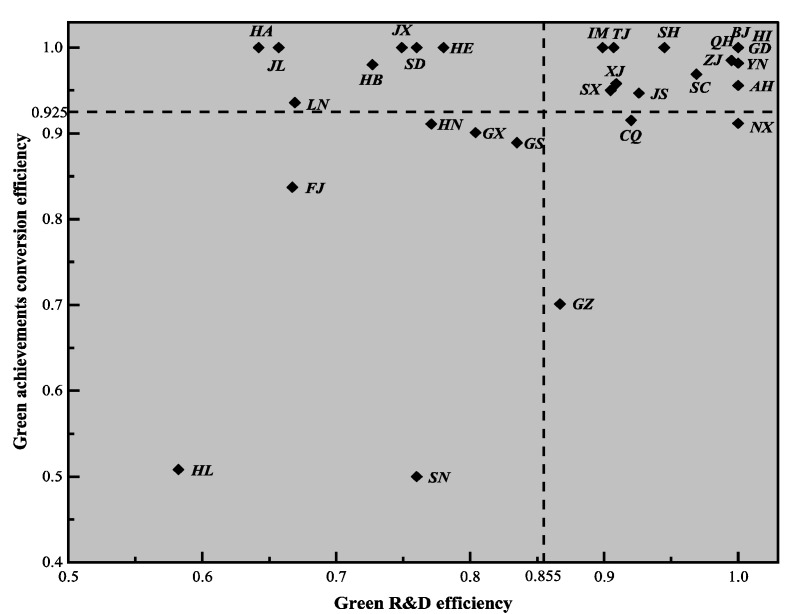
The regional classification of industrial enterprises’ green technology innovation under different combination modes.

**Figure 11 ijerph-16-00940-f011:**
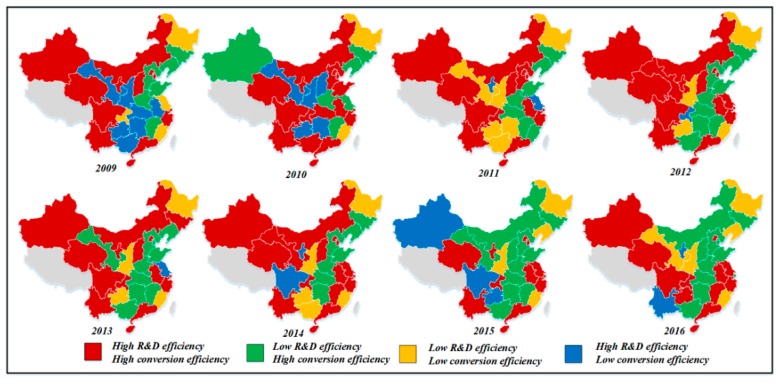
The evolution map of the classification of industrial enterprises’ green innovation in China (2009–2016).

**Table 1 ijerph-16-00940-t001:** The two stages of industrial enterprises’ green innovation indicators with shared input.

Primary Indicators	Secondary Indicators
Green technology innovation input	R&D personnel full-time equivalent
	R&D intramural expenditure capital stock
	Technology acquisition and assimilation expenditure capital stock
	New product development expenditure capital stock
Green technology innovation mid product	Patent applications
	Inventions in force
	New products
Green technology innovation output	Reduced sales revenue of new products
	Reduced revenue for principle business
	Industrial carbon emissions of per unit GDP
	Waste gas, sewage, and general industrial solid-waste emissions of per unit GDP

**Table 2 ijerph-16-00940-t002:** Number of industrial enterprises in various provinces in China (2009–2016).

Area	Number of Industrial Enterprises							
	2009	2010	2011	2012	2013	2014	2015	2016
Entire Country	434,274	452,775	325,553	343,705	352,476	377,791	383,044	378,491
East Area	283,360	291,092	196,949	204,646	207,751	222,012	224,235	219,540
Central Area	71,127	77,923	64,026	70,099	73,198	82,092	86,160	88,678
West Area	46,079	49,151	39,129	42,416	44,515	48,364	50,501	52,299
North-East Area	33,708	34,609	25,449	26,544	27,012	25,323	22,148	17,974
Beijing (BJ)	6890	6884	3746	3692	3701	3686	3548	3340
Tianjin (TJ)	8326	7947	5013	5342	5383	5501	5525	5203
Hebei (HE)	13,096	13,927	11,570	12,360	12,649	14,792	15,295	14,764
Shanxi (SX)	4023	4240	3675	3905	3946	3906	3845	3548
Inner Mongolia (IM)	4465	4611	4175	4244	4377	4413	4404	4289
Liaoning (LN)	23,364	23,832	16,914	17,347	17,561	15,707	12,304	8025
Jilin (JL)	5936	6181	5158	5286	5353	5311	5682	6003
Heilongjiang (HL)	4408	4596	3377	3911	4098	4305	4162	3946
Shanghai (SH)	17,906	16,684	9962	9772	9782	9469	8994	8351
Jiangsu (JS)	60,817	64,136	43,368	45,859	46,387	48,708	48,488	47,900
Zhejiang (ZJ)	59,971	64,364	34,698	36,496	36,904	40,841	41,167	40,128
Anhui (AH)	14,122	16,277	12,432	14,514	15,114	17,762	19,077	19,838
Fujian (FJ)	18,154	19,227	14,116	15,333	15,806	16,744	17,240	17,262
Jiangxi (JX)	7539	7908	6481	7217	7601	8996	9941	10,931
Shandong (SD)	45,518	44,037	35,813	37,625	38,654	40,756	41,485	39,567
Henan (HA)	18,105	19,548	18,328	19,237	19,773	21,748	22,892	23,679
Hubei (HB)	14,027	16,106	10,633	12,441	13,441	15,957	16,413	16,296
Hunan (HN)	13,311	13,844	12,477	12,785	13,323	13,723	13,992	14,386
Guangdong (GD)	52,188	53,389	38,305	37,790	38,094	41,133	42,113	42,688
Guangxi (GX)	5678	6583	5046	5239	5396	5447	5518	5464
Hainan (HI)	494	497	358	377	391	382	380	337
Chongqing (CQ)	6412	7130	4778	4985	5237	6158	6608	6782
Sichuan (SC)	13,267	13,706	12,085	12,719	13,163	13,267	13,525	13,819
Guizhou (GZ)	2791	2963	2329	2752	3139	3895	4482	5123
Yunnan (YN)	3489	3599	2773	3211	3382	3797	3876	4194
Shaanxi (SN)	4480	4564	3684	4284	4489	5081	5413	5862
Gansu (GS)	1987	2000	1371	1735	1830	2091	2148	2105
Qinghai (QH)	523	555	386	423	465	568	575	593
Ningxia (NX)	969	975	764	865	935	1170	1245	1174
Xinjiang (XJ)	2018	2465	1738	1959	2102	2477	2707	2894

**Table 3 ijerph-16-00940-t003:** China’s industrial enterprises’ green technology innovation R&D efficiency (2009–2016).

Area	Green Technology Innovation R&D Efficiency								Mean
	2009	2010	2011	2012	2013	2014	2015	2016	
Entire Country	0.900	0.932	0.866	0.857	0.840	0.844	0.810	0.789	0.855
East Area	0.887	0.951	0.926	0.910	0.874	0.903	0.880	0.851	0.898
Central Area	0.871	0.934	0.814	0.773	0.762	0.787	0.732	0.720	0.799
West Area	0.977	0.978	0.902	0.909	0.894	0.906	0.850	0.832	0.906
North-East Area	0.721	0.692	0.644	0.662	0.681	0.535	0.589	0.562	0.636
Beijing (BJ)	1.000	1.000	1.000	1.000	1.000	1.000	1.000	1.000	1.000
Tianjin (TJ)	1.000	1.000	1.000	0.919	0.806	0.848	0.851	0.766	0.899
Hebei (HE)	0.839	1.000	0.899	0.805	0.664	0.711	0.689	0.635	0.780
Shanxi (SX)	0.952	1.000	1.000	1.000	1.000	0.879	0.769	0.640	0.905
Inner Mongolia (IM)	1.000	1.000	1.000	1.000	1.000	1.000	0.684	0.571	0.907
Liaoning (LN)	0.733	0.709	0.668	0.759	0.624	0.655	0.544	0.656	0.669
Jilin (JL)	0.751	0.469	0.727	0.757	1.000	0.466	0.614	0.473	0.657
Heilongjiang (HL)	0.678	0.897	0.537	0.471	0.419	0.484	0.608	0.558	0.582
Shanghai (SH)	1.000	1.000	1.000	1.000	0.932	1.000	0.848	0.778	0.945
Jiangsu (JS)	0.715	0.873	1.000	0.996	0.945	1.000	1.000	0.882	0.926
Zhejiang (ZJ)	1.000	1.000	1.000	1.000	1.000	1.000	1.000	1.000	1.000
Anhui (AH)	1.000	1.000	1.000	1.000	1.000	1.000	1.000	1.000	1.000
Fujian (FJ)	0.586	0.638	0.642	0.653	0.672	0.665	0.727	0.753	0.667
Jiangxi (JX)	0.659	0.846	0.589	0.614	0.636	0.860	0.790	1.000	0.749
Shandong (SD)	0.727	1.000	0.719	0.728	0.722	0.803	0.680	0.698	0.760
Henan (HA)	0.701	0.811	0.577	0.595	0.600	0.638	0.628	0.582	0.642
Hubei (HB)	0.911	0.947	0.861	0.673	0.619	0.663	0.582	0.563	0.727
Hunan (HN)	1.000	1.000	0.856	0.754	0.717	0.684	0.624	0.532	0.771
Guangdong (GD)	1.000	1.000	1.000	1.000	1.000	1.000	1.000	1.000	1.000
Guangxi (GX)	1.000	0.993	0.849	0.686	0.695	0.681	0.758	0.773	0.804
Hainan (HI)	1.000	1.000	1.000	1.000	1.000	1.000	1.000	1.000	1.000
Chongqing (CQ)	0.821	1.000	0.892	0.858	0.901	0.989	0.960	0.914	0.917
Sichuan (SC)	1.000	1.000	1.000	1.000	1.000	1.000	0.907	0.848	0.969
Guizhou (GZ)	1.000	1.000	0.761	0.739	0.690	0.747	1.000	1.000	0.867
Yunnan (YN)	1.000	1.000	1.000	1.000	1.000	0.961	1.000	1.000	0.995
Shaanxi (SN)	1.000	1.000	0.706	0.847	0.727	0.682	0.566	0.549	0.760
Gansu (GS)	1.000	1.000	0.790	0.903	0.822	0.904	0.647	0.617	0.835
Qinghai (QH)	1.000	1.000	1.000	1.000	1.000	1.000	1.000	1.000	1.000
Ningxia (NX)	1.000	1.000	1.000	1.000	1.000	1.000	1.000	1.000	1.000
Xinjiang (XJ)	0.922	0.763	0.920	0.965	1.000	1.000	0.828	0.876	0.909

**Table 4 ijerph-16-00940-t004:** China’s industrial enterprises’ green technology innovation achievements in transformation efficiency (2009–2016).

Area	Green Technology Innovation Achievements Transformation Efficiency								Mean
	2009	2010	2011	2012	2013	2014	2015	2016	
Entire Country	0.867	0.914	0.913	0.947	0.951	0.934	0.949	0.934	0.926
East Area	0.952	0.998	0.987	0.992	0.970	0.975	0.968	0.971	0.977
Central Area	0.847	0.893	0.988	1.000	1.000	1.000	1.000	1.000	0.966
West Area	0.806	0.872	0.834	0.913	0.934	0.891	0.938	0.922	0.889
North-East Area	0.853	0.829	0.809	0.814	0.847	0.823	0.821	0.723	0.815
Beijing (BJ)	1.000	1.000	1.000	1.000	1.000	1.000	1.000	1.000	1.000
Tianjin (TJ)	1.000	1.000	1.000	1.000	1.000	1.000	1.000	1.000	1.000
Hebei (HE)	1.000	1.000	1.000	1.000	1.000	1.000	1.000	1.000	1.000
Shanxi (SX)	0.968	0.629	1.000	1.000	1.000	1.000	1.000	1.000	0.950
Inner Mongolia (IM)	1.000	1.000	1.000	1.000	1.000	1.000	1.000	1.000	1.000
Liaoning (LN)	0.974	1.000	1.000	1.000	1.000	1.000	0.899	0.617	0.936
Jilin (JL)	1.000	1.000	1.000	1.000	1.000	1.000	1.000	1.000	1.000
Heilongjiang (HL)	0.584	0.486	0.426	0.441	0.541	0.469	0.563	0.551	0.508
Shanghai (SH)	1.000	1.000	1.000	1.000	1.000	1.000	1.000	1.000	1.000
Jiangsu (JS)	0.860	0.976	0.902	0.974	0.906	0.985	0.971	1.000	0.947
Zhejiang (ZJ)	0.883	0.999	0.970	1.000	1.000	1.000	1.000	1.000	0.982
Anhui (AH)	0.695	0.950	1.000	1.000	1.000	1.000	1.000	1.000	0.956
Fujian (FJ)	0.777	1.000	1.000	0.943	0.795	0.762	0.711	0.709	0.837
Jiangxi (JX)	1.000	1.000	1.000	1.000	1.000	1.000	1.000	1.000	1.000
Shandong (SD)	1.000	1.000	1.000	1.000	1.000	1.000	1.000	1.000	1.000
Henan (HA)	1.000	1.000	1.000	1.000	1.000	1.000	1.000	1.000	1.000
Hubei (HB)	0.839	1.000	1.000	1.000	1.000	1.000	1.000	1.000	0.980
Hunan (HN)	0.582	0.780	0.928	1.000	1.000	1.000	1.000	1.000	0.911
Guangdong (GD)	1.000	1.000	1.000	1.000	1.000	1.000	1.000	1.000	1.000
Guangxi (GX)	0.709	0.971	0.705	0.998	1.000	0.823	1.000	1.000	0.901
Hainan (HI)	1.000	1.000	1.000	1.000	1.000	1.000	1.000	1.000	1.000
Chongqing (CQ)	0.485	1.000	1.000	0.941	1.000	1.000	1.000	1.000	0.928
Sichuan (SC)	0.954	0.985	1.000	1.000	1.000	0.930	0.885	1.000	0.969
Guizhou (GZ)	0.551	0.527	0.556	0.591	0.718	0.721	0.941	1.000	0.701
Yunnan (YN)	1.000	1.000	1.000	1.000	1.000	1.000	1.000	0.878	0.985
Shaanxi (SN)	0.511	0.507	0.487	0.509	0.558	0.433	0.493	0.503	0.500
Gansu (GS)	0.719	0.682	0.782	1.000	1.000	1.000	1.000	0.928	0.889
Qinghai (QH)	1.000	1.000	1.000	1.000	1.000	1.000	1.000	1.000	1.000
Ningxia (NX)	0.950	1.000	0.719	1.000	1.000	0.899	1.000	0.837	0.926
Xinjiang (XJ)	0.983	0.925	0.928	1.000	1.000	1.000	0.828	1.000	0.958

**Table 5 ijerph-16-00940-t005:** China’s industrial enterprises’ green technology innovation comprehensive efficiency (2009–2016).

Area	Green Technology Innovation Comprehensive Efficiency								Mean
	2009	2010	2011	2012	2013	2014	2015	2016	
Entire Country	0.781	0.851	0.791	0.812	0.798	0.788	0.769	0.737	0.791
East Area	0.844	0.949	0.914	0.903	0.848	0.880	0.852	0.826	0.877
Central Area	0.738	0.834	0.804	0.773	0.762	0.787	0.732	0.720	0.769
West Area	0.787	0.853	0.752	0.830	0.835	0.807	0.797	0.767	0.804
North-East Area	0.614	0.573	0.521	0.539	0.577	0.440	0.483	0.406	0.519
Beijing (BJ)	1.000	1.000	1.000	1.000	1.000	1.000	1.000	1.000	1.000
Tianjin (TJ)	1.000	1.000	1.000	0.919	0.806	0.848	0.851	0.766	0.899
Hebei (HE)	0.839	1.000	0.899	0.805	0.664	0.711	0.689	0.635	0.780
Shanxi (SX)	0.922	0.629	1.000	1.000	1.000	0.879	0.769	0.640	0.855
Inner Mongolia (IM)	1.000	1.000	1.000	1.000	1.000	1.000	0.684	0.571	0.907
Liaoning (LN)	0.714	0.709	0.668	0.759	0.624	0.655	0.489	0.405	0.628
Jilin (JL)	0.751	0.469	0.727	0.757	1.000	0.466	0.614	0.473	0.657
Heilongjiang (HL)	0.396	0.436	0.229	0.208	0.227	0.227	0.342	0.307	0.296
Shanghai (SH)	1.000	1.000	1.000	1.000	0.932	1.000	0.848	0.778	0.945
Jiangsu (JS)	0.615	0.852	0.902	0.970	0.856	0.985	0.971	0.882	0.879
Zhejiang (ZJ)	0.883	0.999	0.970	1.000	1.000	1.000	1.000	1.000	0.982
Anhui (AH)	0.695	0.950	1.000	1.000	1.000	1.000	1.000	1.000	0.956
Fujian (FJ)	0.455	0.638	0.642	0.616	0.534	0.507	0.517	0.534	0.555
Jiangxi (JX)	0.659	0.846	0.589	0.614	0.636	0.860	0.790	1.000	0.749
Shandong (SD)	0.727	1.000	0.719	0.728	0.722	0.803	0.680	0.698	0.760
Henan (HA)	0.701	0.811	0.577	0.595	0.600	0.638	0.628	0.582	0.642
Hubei (HB)	0.764	0.947	0.861	0.673	0.619	0.663	0.582	0.563	0.709
Hunan (HN)	0.582	0.780	0.794	0.754	0.717	0.684	0.624	0.532	0.683
Guangdong (GD)	1.000	1.000	1.000	1.000	1.000	1.000	1.000	1.000	1.000
Guangxi (GX)	0.709	0.964	0.599	0.685	0.695	0.560	0.758	0.773	0.718
Hainan (HI)	1.000	1.000	1.000	1.000	1.000	1.000	1.000	1.000	1.000
Chongqing (CQ)	0.398	1.000	0.892	0.807	0.901	0.989	0.960	0.914	0.858
Sichuan (SC)	0.954	0.985	1.000	1.000	1.000	0.930	0.803	0.848	0.940
Guizhou (GZ)	0.551	0.527	0.423	0.437	0.495	0.539	0.941	1.000	0.614
Yunnan (YN)	1.000	1.000	1.000	1.000	1.000	0.961	1.000	0.878	0.980
Shaanxi (SN)	0.511	0.507	0.344	0.431	0.406	0.295	0.279	0.276	0.381
Gansu (GS)	0.719	0.682	0.618	0.903	0.822	0.904	0.647	0.573	0.733
Qinghai (QH)	1.000	1.000	1.000	1.000	1.000	1.000	1.000	1.000	1.000
Ningxia (NX)	0.950	1.000	0.719	1.000	1.000	0.899	1.000	0.837	0.926
Xinjiang (XJ)	0.906	0.706	0.854	0.965	1.000	1.000	0.828	0.876	0.892

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
