# Peer review of "Assessing Regional Differences in Green Innovation Efficiency of Industrial Enterprises in China"

_ijerph, 2019, doi:10.3390/ijerph16060940_

Round 1

Reviewer 1 Report

Good paper. However, the conclusion and suggestions should be reduced.

Author Response

1. Good paper. However, the conclusion and suggestions should be reduced.

Thank you for your positive comments. To address your concerns, some conclusion and suggestions have been reduced in this revision. The details are as follows.

“From the perspective of enterprises, enterprises should rationally allocate talent, information, capital and other investment factors, pay attention to the sharing of green innovation resources in two stage, and strengthen the interaction between the two links. Specifically, enterprises should focus on the management of green patent output, technology incubation, especially the original green technology development activities from 0 to 1. Regarding enterprises as the main body of green technology innovation is not to deny the important role of research institutes and universities in technological innovation, especially its important leading role in technological breakthroughs. Larger companies need to further streamline and optimize their organizational structure, increase sensitivity to new markets and new products, and strengthen investment in green technology development and clean production equipment. The performance appraisal of state-owned enterprises can not only focus on the number of patents, but also pay attention to the quality of patents, and reward enterprises and employees who realize green and commercial technology. The good functioning of a system requires the parties involved to work together around a goal. In the future, enterprises in the eastern region must vigorously carry out original innovation, introduction the innovation and integrated innovation of digestion and absorption. Enterprises need to work hard to speed up the construction of modern enterprise systems and management reformation, increase the endogenous power of enterprises, and pay attention to the improvement of management level of green technology innovation. By doing this, these enterprises can drive the central and western regions to shift to a high-quality, low-pollution economic growth mode. Central and western enterprises need to strengthen cooperation with eastern enterprises and implement pairing assistance. At the same time, central and western enterprises need to introduce high-tech talents, pay attention to the marketization and commercialization of technology, and improve the volume and efficiency of green R&D and achievements transformation.

Strengthen the construction of green technology innovation culture. Innovative cultural environment is very important for scientific and technological innovation. It can not only stimulate the enthusiasm of employees, but also enable R&D personnel to share common goals and achieve their goals. Integrating innovation into science and technology workers’ daily and work life and forming a good innovative cultural environment that dares to innovate and tolerate failure. At the same time, we must respect the regular of innovation and establish a scientific and rational innovation evaluation mechanism. In addition, promoting institution construction and management innovation can enhance the enthusiasm of overall scientific and technological innovation, improve the output efficiency of scientific and technological achievements, and ensure the smooth progress of the scientific and technological innovation process.

Improve the industry-university-research cooperation innovation system. It is necessary to speed up the establishment of an enterprise-led industry-university-research R&D mechanism, and form an innovation system that market-led, enterprise-oriented and combines industry-university-research. At the same time, further improve the benefits distribution and risk sharing mechanism of industry-university-research cooperation. By continuously improving the innovation mechanism of industry-university-research cooperation, we will make good use of the advantages and subjective initiative of innovation entities such as enterprises, universities and scientific research institutions, and improve the level of integrated green innovation.

Optimize the green innovation policy to ensure the environment. On the basis of increasing innovation support and preferential efforts, we should establish a diversified investment and financing system and risk sharing mechanism, increase the implementation of scientific and technological innovation policies, and establish good public opinion atmosphere and social culture that supports and respects innovation. The government should formulate relevant laws and regulations on intellectual property protection, technology intermediary services, transformation of scientific and technological achievements, and construction of integrated innovation platforms. And provide good policy guarantee environment for innovation activities by improving the science and technology innovation laws and systems.”

Reviewer 2 Report

See attachment.

Author Response

1. Weak of contributions.

Thank you for your suggestions. I am sorry that our article description has caused you trouble, please allow us to explain. Our article belongs to the application category, focusing on the analysis of Chinese enterprises' green technology innovation issues and providing relevant countermeasures and suggestions for the government. We have already shown the contribution of this article in Section 1 in this revision, which are shown as follows.

“In this paper, with respect to the problems of regional difference of China’s industrial enterprises green innovation efficiency, from the perspective of two-stage innovation value chain, taking China’s industrial enterprises as the research objects, we establish a novel green innovation efficiency evaluation index system of industrial enterprises by introducing the industrial carbon emissions of per unit GDP and the “three wastes” pollutants into the research framework of green technology innovation efficiency, and then exploit a two-stage network DEA with the shared input to measure the efficiency of regional enterprises green technology innovation and explore the efficiency of regional differences in industrial enterprises green technology R&D and green technology achievement transformation.”

2. Introduction.

Thank you for your forethoughtful suggestions. With your comments in mind, we rewrote the introduction. in this revision. The details are as follows.

“The severe climate change and environmental pollution problems have been attracting more and more countries’ concern and attention, so that how to transform their traditional economic growth patterns, effectively improve the environment quality under maintaining the original level of economic development and make good use of resources and achieve sustainable development has become an much more important issue for many countries. Being an important mean to break out the constraints of resources and the environment, enhance the competitiveness of enterprises, and achieve the upgrading of industrial structure and promote high-quality economic growth, green technology innovation has been accepting by more and more countries [1]. Innovation efficiency reflects the company's full use of the level of innovation resources, and has a profound impact on the formation of competitive advantage [2]. Facing to severe resource and environmental problems, to improve national or regional competitiveness and achieve the healthy development of national or regional economic, Brawn and Wield (1994) proposed the concept of green technology innovation green technology innovation [3], and then some scholars made research on green technology innovation efficiency from the perspectives of natural resource [4], institutional theory [5], social network relationship [6], stakeholders [7] and CSR [8], respectively. To sum up, these researches mainly focused on the aspects of defining the conception of green technology innovation, analyzing the interactions between green technology innovation, environmental rules and social economic [9,10], identifying the factors affecting green technology innovation [11-13], constructing an evaluation index system for green technology innovation efficiency, and measuring the efficiency of green technology innovation. In 2017, Thomson Reuters announced the list of the top 100 innovative enterprises in the world, no mainland Chinese companies were selected, and the technological innovation output and efficiency of industrial enterprises still need to be improved. In addition, with the growth of China's industrial economy, the environmental pollution problem is becoming more and more serious. The contradiction between the science and technology economy and the ecological environment has become increasingly prominent, and people's production activities and life have been greatly affected. However, the current research on the measurement of green technology innovation efficiency of China's provincial industrial enterprises is not complete enough and does not fully consider the phased and shared investment relevance of green technology innovation efficiency, the evaluation index system is not reasonable and perfect, which leads to the failure to fully exploit the differences and characteristics of provincial innovation efficiency. Therefore, the indicator system and measurement method are the basis for measuring the efficiency of green technology innovation. In this paper, with respect to the problems of regional difference of China’s industrial enterprises green innovation efficiency, from the perspective of two-stage innovation value chain, taking China’s industrial enterprises as the research objects, we establish a novel green innovation efficiency evaluation index system of industrial enterprises by introducing the industrial carbon emissions of per unit GDP and the “three wastes” pollutants into the research framework of green technology innovation efficiency, and then exploit a two-stage network DEA with the shared input to measure the efficiency of regional enterprises green technology innovation and explore the efficiency of regional differences in industrial enterprises green technology R&D and green technology achievement transformation.

The remainder of this paper is organized as follows. In Section 2, we give the related literature review on green technology innovation. In Section 3, a novel green technology innovation efficiency evaluation index system of industrial enterprises is designed. Section 4 stablishes a two-stage network DEA model with shared related input. In Section 5, we exploit the two-stage network DEA model to measure regional industrial enterprises green innovation efficiency, and analyze their differences. Section 6 concludes the paper with some remarks and gives some suggestions”

3. Literature review

We are grateful for your insightful comments. To address your concerns, we have revised and refined the literature review in this revision. The details are as follows.

“According to the ecological-economic development needs, we should find the relationship among rapid economics development, resources excessive utilization and natural environmental degradation, analyze the impact factors and improve the ecological-economic efficiency of green technology innovation [14-16]. The measurement and evaluation of green technology innovation efficiency involves multi-stage, multi-angle and multi-factors, which is a complex system engineering. Through the existing research results, it can be summarized as the following three aspects: the identification of factors and bottlenecks of green technology innovation efficiency, the design of green technology innovation efficiency evaluation index system, and the construction of green technology innovation efficiency measurement methods.

At present, the analysis of the factors affecting the efficiency of green technology innovation technology is mainly concentrated on the two levels of macro policy, and industrial enterprise. In terms of macroeconomic policies, scholars' research mainly reveals the importance of government support [17,18], policy portfolio [19], and environmental rules [9] for green technology innovation. For industrial enterprises themselves, they should focus on corporate governance [13], green investment [20,21], social reciprocity[6] and other aspects.

To make assessment of green technology innovation efficiency, a reasonable index system should be established. The existing research on green technology innovation efficiency evaluation index system is mainly to integrate environmental factors and technological innovation efficiency indicators, and to construct a new green performance evaluation index system. From the four aspects of management innovation, process innovation, product innovation, technological innovation, Tseng et al. [22] discussed green innovation, and then constructed the enterprise green technology innovation efficiency evaluation system including 22 indexes of investment in green equipment and technology, implementation of comprehensive material saving plan, supervision system and technology transfer, advanced green production technology, management of documentation and information. Luo and Liang [23] constructed the green technology innovation efficiency evaluation system of regional industrial enterprises in China from the perspective of green technology innovation input, intermediate output, expected output and undesired output. Although these indicators are comprehensive, their inherent logical structure is unreasonable, and it is difficult to effectively evaluate the efficiency of green technology innovation.”

4. Index

Thank you for your positive comments. To address your concerns, we further explain the rationality of the indexes and add some citations to make it more reasonable. The details are as follows.

“R&D investment, which is mainly measured by R&D expenses and R&D personnel, is an important form of innovation resources. Relevant research shows that R&D investment is closely related to green innovation efficiency [36].

The full-time equivalent of R&D personnel (it refers to the sum of the workload of R&D full-time personnel and the workload of part-time personnel converted according to actual working hours.), the stock of R&D expenditure, the stock of imported digestion and absorption expenses (it refers to the work carried out on the mastery, application and reproduction of imported technologies, as well as innovations based on them.)

Where, the stock refers to the balance of products, goods, reserves, assets and liabilities that were produced and accumulated in the past at a specified time.

Intermediate output. The intermediate output of technological innovation in industrial enterprises is generally reflected in patents and new product development projects [37,38]. Some studies have shown that patents play a role in the development and diffusion of green technologies[39].

the number of valid invention patents (it refers to a patent that is still in a valid state after the patent application is authorized)

Based on the above analysis and discussion, with reference to relevant literatures [18,26,34,37], in this paper, the full-time equivalent of R&D personnel, the stock of R&D expenditure, the stock of imported digestion and absorption expenses and the stock of new product development funds are chosen as the indexes of industrial enterprises technological innovation input, and the number of patent applications, the number of valid invention patents and the number of new product development projects are taken as intermediate output indexes, while new product sales revenue and main business income are determined as economic output indexes, and industrial carbon emissions per unit of industrial GDP and industrial “three wastes” pollutants per unit of industrial GDP are considered as final environmental output indexes. This indicator system completely describes the whole process of industrial enterprise green technology innovation (from the previous capital, personnel input to the medium-term patent output, the later green product output), fully considering the economic and environmental factors. So that we establish a two-stage efficiency evaluation index system for industrial enterprises green technology innovation shown in Table 1.”

5. Results

I am sorry that our article description has caused you trouble, please allow us to explain. Tables 2-4 show R&D efficiency, efficiency of results conversion, and overall efficiency. Figures 7-9 show the three efficiency changes over the observation period.

6. Quality of communication

We are very sorry to say that we have made a few grammatical mistakes and we have changed it, which can be seen in this revision. Thank you very much for your pointing out the error.

Reviewer 3 Report

This paper is interesting and of great practical value. The author's research ideas are clear, and the application of research methods is reasonable. However, there are still some problems that need further clarification and to be modified. Specific Suggestions are as follows:

Major issues:

Some research conclusions are not convincing, and the following key issues need to be solved:

1. The biggest problem lies in the measurement of key variables, such as Green technology innovation investment and the number of patents. How can these variables be guaranteed to be related to environmental protection or Green-technology? According to the description of the paper, it appears to refer to all investment and patents.

2. In addition, due to the severe imbalance in China's economic development, the number of industrial enterprises in some backward areas in western China (Qinghai, Ningxia, Xinjiang provinces) is far lower than that in economically developed areas, and relevant variables of national policies, economy, and technological progress cannot be ignored. If these factors are not taken into account in the model, there will be a large bias in the research results, which will result in some inconsistent with previous studies. As the author has mentioned in many places:

Line: 374 ". The analysis results can not completely cooperate with The previous research results...";

Line: 438 "...Will be completely in line with the previous research results...";

Line:600-601 "The efficiency of green technology innovation is not completely consistent with The previous research results...".

In order to improve the credibility and validity of the paper's conclusions, I think the author carefully examines the possible influence of these neglected variables on the research conclusions.

3. The author should provide the number of industrial enterprises in each region and province, as well as descriptive information of relevant indicators, which is crucial for readers to judge the conclusion.

Other comments:

1. The Introduction part needs to be revised and adjusted to reflect a good narrative style to introduce the research questions.

2. The logic of Literature Review is not clear, and the author is suggested to summarize the factors related to green innovation from different levels or aspects such as enterprises, industries, regions, and macro policies.

3. The Index in the Evaluation Index System of Table 1 lacks the support from source literatures, and the author should supplement relevant researches to prove the rationality of the Index.

Author Response

Response to Reviewer #3

Major issues:

1. The biggest problem lies in the measurement of key variables, such as Green technology innovation investment and the number of patents. How can these variables be guaranteed to be related to environmental protection or Green-technology? According to the description of the paper, it appears to refer to all investment and patents.

Thank you for your positive comments. To address your concerns, we further explain the rationality of the indexes and add some citations to make it more reasonable. The details are as follows.

“R&D investment, which is mainly measured by R&D expenses and R&D personnel, is an important form of innovation resources. Relevant research shows that R&D investment is closely related to green innovation efficiency [36].

The full-time equivalent of R&D personnel (it refers to the sum of the workload of R&D full-time personnel and the workload of part-time personnel converted according to actual working hours.), the stock of R&D expenditure, the stock of imported digestion and absorption expenses (it refers to the work carried out on the mastery, application and reproduction of imported technologies, as well as innovations based on them.)

Where, the stock refers to the balance of products, goods, reserves, assets and liabilities that were produced and accumulated in the past at a specified time.

Intermediate output. The intermediate output of technological innovation in industrial enterprises is generally reflected in patents and new product development projects [37,38]. Some studies have shown that patents play a role in the development and diffusion of green technologies[39].

the number of valid invention patents (it refers to a patent that is still in a valid state after the patent application is authorized)

Based on the above analysis and discussion, with reference to relevant literatures [18,26,34,37], in this paper, the full-time equivalent of R&D personnel, the stock of R&D expenditure, the stock of imported digestion and absorption expenses and the stock of new product development funds are chosen as the indexes of industrial enterprises technological innovation input, and the number of patent applications, the number of valid invention patents and the number of new product development projects are taken as intermediate output indexes, while new product sales revenue and main business income are determined as economic output indexes, and industrial carbon emissions per unit of industrial GDP and industrial “three wastes” pollutants per unit of industrial GDP are considered as final environmental output indexes. This indicator system completely describes the whole process of industrial enterprise green technology innovation (from the previous capital, personnel input to the medium-term patent output, the later green product output), fully considering the economic and environmental factors. So that we establish a two-stage efficiency evaluation index system for industrial enterprises green technology innovation shown in Table 1.”

2. In addition, due to the severe imbalance in China's economic development, the number of industrial enterprises in some backward areas in western China (Qinghai, Ningxia, Xinjiang provinces) is far lower than that in economically developed areas, and relevant variables of national policies, economy, and technological progress cannot be ignored. If these factors are not taken into account in the model, there will be a large bias in the research results, which will result in some inconsistent with previous studies.

Thank you for your insightful comments. The results we obtained are consistent with recent research in some literature. We give some citations and further explain the reasons for this result. The details are as follows.

“Since this paper incorporates industrial carbon emissions per unit of GDP and “three wastes” pollutants into the performance evaluation index system, and uses the shared input-related two-stage network DEA model to measure, some problems that have been neglected in previous studies are found. The analysis results we concluded above are not completely consistent with the previous research results: “east, middle and west's efficiency regular descending”, some literature studies have reached similar conclusions [34,36]. Although there are a small number of industrial enterprises in the western region, they have more national key laboratories, scientific research institutions, and national key support policies and funds, while the degree of environmental damage is small [34,37]. The eastern region has developed economy and high level of science and technology. The introduction of foreign-funded enterprises has also brought about certain technological demonstrations and spillover effects, and the innovation environment and marketization process have been continuously improved. Therefore, enterprises in these regions have achieved intermediate output such as large patents and new product development projects under the input of established research and development funds and personnel, making the green research and development efficiency at a better level in the region.”

3. The author should provide the number of industrial enterprises in each region and province, as well as descriptive information of relevant indicators, which is crucial for readers to judge the conclusion.

Thank you for your insightful comments. We have added the number of industrial enterprises in each region and province, as well as descriptive information of relevant indicators in this revision. The details are as follows.

In addition, in order to fully demonstrate the problems we have studied, we give the number of industrial enterprises in various provinces in China during the observation period, as shown in Table 2.

Table 2. Number of industrial enterprises in various provinces in China (2009–2016).

Area

Number of industrial enterprises

2009

2010

2011

2012

2013

2014

2015

2016

Entire Country

434274

452775

325553

343705

352476

377791

383044

378491

East Area

283360

291092

196949

204646

207751

222012

224235

219540

Central Area

71127

77923

64026

70099

73198

82092

86160

88678

West Area

46079

49151

39129

42416

44515

48364

50501

52299

North-east Area

33708

34609

25449

26544

27012

25323

22148

17974

Beijing (BJ)

6890

6884

3746

3692

3701

3686

3548

3340

Tianjin (TJ)

8326

7947

5013

5342

5383

5501

5525

5203

Hebei (HE)

13096

13927

11570

12360

12649

14792

15295

14764

Shanxi (SX)

4023

4240

3675

3905

3946

3906

3845

3548

Inner Mongoria (IM)

4465

4611

4175

4244

4377

4413

4404

4289

Liaoning (LN)

23364

23832

16914

17347

17561

15707

12304

8025

Jilin (JL)

5936

6181

5158

5286

5353

5311

5682

6003

Heilongjiang (HL)

4408

4596

3377

3911

4098

4305

4162

3946

Shanghai (SH)

17906

16684

9962

9772

9782

9469

8994

8351

Jiangsu (JS)

60817

64136

43368

45859

46387

48708

48488

47900

Zhejiang (ZJ)

59971

64364

34698

36496

36904

40841

41167

40128

Anhui (AH)

14122

16277

12432

14514

15114

17762

19077

19838

Fujian (FJ)

18154

19227

14116

15333

15806

16744

17240

17262

Jiangxi (JX)

7539

7908

6481

7217

7601

8996

9941

10931

Shandong (SD)

45518

44037

35813

37625

38654

40756

41485

39567

Henan (HA)

18105

19548

18328

19237

19773

21748

22892

23679

Hubei (HB)

14027

16106

10633

12441

13441

15957

16413

16296

Hunan (HN)

13311

13844

12477

12785

13323

13723

13992

14386

Guangdong (GD)

52188

53389

38305

37790

38094

41133

42113

42688

Guangxi (GX)

5678

6583

5046

5239

5396

5447

5518

5464

Hainan (HI)

494

497

358

377

391

382

380

337

Chongqing (CQ)

6412

7130

4778

4985

5237

6158

6608

6782

Sichuan (SC)

13267

13706

12085

12719

13163

13267

13525

13819

Guizhou (GZ)

2791

2963

2329

2752

3139

3895

4482

5123

Yunnan (YN)

3489

3599

2773

3211

3382

3797

3876

4194

Shaanxi (SN)

4480

4564

3684

4284

4489

5081

5413

5862

Gansu (GS)

1987

2000

1371

1735

1830

2091

2148

2105

Qinghai (QH)

523

555

386

423

465

568

575

593

Ningxia (NX)

969

975

764

865

935

1170

1245

1174

Xinjiang (XJ)

2018

2465

1738

1959

2102

2477

2707

2894

R&D investment, which is mainly measured by R&D expenses and R&D personnel, is an important form of innovation resources. Relevant research shows that R&D investment is closely related to green innovation efficiency [36].

The full-time equivalent of R&D personnel (it refers to the sum of the workload of R&D full-time personnel and the workload of part-time personnel converted according to actual working hours.), the stock of R&D expenditure, the stock of imported digestion and absorption expenses (it refers to the work carried out on the mastery, application and reproduction of imported technologies, as well as innovations based on them.)….. Where, the stock refers to the balance of products, goods, reserves, assets and liabilities that were produced and accumulated in the past at a specified time.

the number of valid invention patents (it refers to a patent that is still in a valid state after the patent application is authorized)

Other comments:

1. The Introduction part needs to be revised and adjusted to reflect a good narrative style to introduce the research questions.

Thank you for your forethoughtful suggestions. With your comments in mind, we rewrote the introduction. in this revision. The details are as follows.

“The severe climate change and environmental pollution problems have been attracting more and more countries’ concern and attention, so that how to transform their traditional economic growth patterns, effectively improve the environment quality under maintaining the original level of economic development and make good use of resources and achieve sustainable development has become an much more important issue for many countries. Being an important mean to break out the constraints of resources and the environment, enhance the competitiveness of enterprises, and achieve the upgrading of industrial structure and promote high-quality economic growth, green technology innovation has been accepting by more and more countries [1]. Innovation efficiency reflects the company's full use of the level of innovation resources, and has a profound impact on the formation of competitive advantage [2]. Facing to severe resource and environmental problems, to improve national or regional competitiveness and achieve the healthy development of national or regional economic, Brawn and Wield (1994) proposed the concept of green technology innovation green technology innovation [3], and then some scholars made research on green technology innovation efficiency from the perspectives of natural resource [4], institutional theory [5], social network relationship [6], stakeholders [7] and CSR [8], respectively. To sum up, these researches mainly focused on the aspects of defining the conception of green technology innovation, analyzing the interactions between green technology innovation, environmental rules and social economic [9,10], identifying the factors affecting green technology innovation [11-13], constructing an evaluation index system for green technology innovation efficiency, and measuring the efficiency of green technology innovation. In 2017, Thomson Reuters announced the list of the top 100 innovative enterprises in the world, no mainland Chinese companies were selected, and the technological innovation output and efficiency of industrial enterprises still need to be improved. In addition, with the growth of China's industrial economy, the environmental pollution problem is becoming more and more serious. The contradiction between the science and technology economy and the ecological environment has become increasingly prominent, and people's production activities and life have been greatly affected. However, the current research on the measurement of green technology innovation efficiency of China's provincial industrial enterprises is not complete enough and does not fully consider the phased and shared investment relevance of green technology innovation efficiency, the evaluation index system is not reasonable and perfect, which leads to the failure to fully exploit the differences and characteristics of provincial innovation efficiency. Therefore, the indicator system and measurement method are the basis for measuring the efficiency of green technology innovation. In this paper, with respect to the problems of regional difference of China’s industrial enterprises green innovation efficiency, from the perspective of two-stage innovation value chain, taking China’s industrial enterprises as the research objects, we establish a novel green innovation efficiency evaluation index system of industrial enterprises by introducing the industrial carbon emissions of per unit GDP and the “three wastes” pollutants into the research framework of green technology innovation efficiency, and then exploit a two-stage network DEA with the shared input to measure the efficiency of regional enterprises green technology innovation and explore the efficiency of regional differences in industrial enterprises green technology R&D and green technology achievement transformation.

The remainder of this paper is organized as follows. In Section 2, we give the related literature review on green technology innovation. In Section 3, a novel green technology innovation efficiency evaluation index system of industrial enterprises is designed. Section 4 stablishes a two-stage network DEA model with shared related input. In Section 5, we exploit the two-stage network DEA model to measure regional industrial enterprises green innovation efficiency, and analyze their differences. Section 6 concludes the paper with some remarks and gives some suggestions”

2. The logic of Literature Review is not clear, and the author is suggested to summarize the factors related to green innovation from different levels or aspects such as enterprises, industries, regions, and macro policies.

We are grateful for your insightful comments. To address your concerns, we have revised and refined the literature review in this revision. The details are as follows

 “According to the ecological-economic development needs, we should find the relationship among rapid economics development, resources excessive utilization and natural environmental degradation, analyze the impact factors and improve the ecological-economic efficiency of green technology innovation [14-16]. The measurement and evaluation of green technology innovation efficiency involves multi-stage, multi-angle and multi-factors, which is a complex system engineering. Through the existing research results, it can be summarized as the following three aspects: the identification of factors and bottlenecks of green technology innovation efficiency, the design of green technology innovation efficiency evaluation index system, and the construction of green technology innovation efficiency measurement methods.

At present, the analysis of the factors affecting the efficiency of green technology innovation technology is mainly concentrated on the two levels of macro policy, and industrial enterprise. In terms of macroeconomic policies, scholars' research mainly reveals the importance of government support [17,18], policy portfolio [19], and environmental rules [9] for green technology innovation. For industrial enterprises themselves, they should focus on corporate governance [13], green investment [20,21], social reciprocity[6] and other aspects.

To make assessment of green technology innovation efficiency, a reasonable index system should be established. The existing research on green technology innovation efficiency evaluation index system is mainly to integrate environmental factors and technological innovation efficiency indicators, and to construct a new green performance evaluation index system. From the four aspects of management innovation, process innovation, product innovation, technological innovation, Tseng et al. [22] discussed green innovation, and then constructed the enterprise green technology innovation efficiency evaluation system including 22 indexes of investment in green equipment and technology, implementation of comprehensive material saving plan, supervision system and technology transfer, advanced green production technology, management of documentation and information. Luo and Liang [23] constructed the green technology innovation efficiency evaluation system of regional industrial enterprises in China from the perspective of green technology innovation input, intermediate output, expected output and undesired output. Although these indicators are comprehensive, their inherent logical structure is unreasonable, and it is difficult to effectively evaluate the efficiency of green technology innovation. According to the innovation value chain [24], green technology innovation generally includes two stages of green technology R&D and green technology transformation. However, the existing researches mostly regard it as a single stage or the whole stage to evaluate the efficiency of green technology innovation, few divide it into two stages to discuss [25]. ”

3. The Index in the Evaluation Index System of Table 1 lacks the support from source literatures, and the author should supplement relevant researches to prove the rationality of the Index.

Thank you for your positive comments. To address your concerns, we further explain the rationality of the indexes and add some citations to make it more reasonable. The details are as follows.

Based on the above analysis and discussion, with reference to relevant literatures [18,26,34,37], in this paper, the full-time equivalent of R&D personnel, the stock of R&D expenditure, the stock of imported digestion and absorption expenses and the stock of new product development funds are chosen as the indexes of industrial enterprises technological innovation input, and the number of patent applications, the number of valid invention patents and the number of new product development projects are taken as intermediate output indexes, while new product sales revenue and main business income are determined as economic output indexes, and industrial carbon emissions per unit of industrial GDP and industrial “three wastes” pollutants per unit of industrial GDP are considered as final environmental output indexes. This indicator system completely describes the whole process of industrial enterprise green technology innovation (from the previous capital, personnel input to the medium-term patent output, the later green product output), fully considering the economic and environmental factors. So that we establish a two-stage efficiency evaluation index system for industrial enterprises green technology innovation shown in Table 1.

R&D investment, which is mainly measured by R&D expenses and R&D personnel, is an important form of innovation resources. Relevant research shows that R&D investment is closely related to green innovation efficiency [36].

Intermediate output. The intermediate output of technological innovation in industrial enterprises is generally reflected in patents and new product development projects [37,38]. Some studies have shown that patents play a role in the development and diffusion of green technologies[39]

Round 2

Reviewer 3 Report

I am satisfied with the revision.The minor suggestion is that the author should supplement some limitations of the paper and future research directions.

Author Response

We deeply appreciate the time and effort you have spent in reviewing our manuscript (ID: ijerph-436218).Your comments are really thoughtful and helpful, we have carefully taken your kind advices and referee's detailed suggestions into consideration in revising our manuscript. Enclosure is our point-point answer to the referee’s comments. We sincerely hope this revised manuscript will be finally acceptable to be published on IJERPH 2019. Thank you very much for all your helps and looking forward to hearing from you soon.